# Age-related delay in visual and auditory evoked responses is mediated by white- and grey-matter differences

D. Price[1], L.K. Tyler[2], R. Neto Henriques[1], K.L. Campbell[3], N. Williams[4], M.S. Treder[2], J.R. Taylor[5], Cam-CAN[†] & R.N.A. Henson[1]

Slowing is a common feature of ageing, yet a direct relationship between neural slowing and brain atrophy is yet to be established in healthy humans. We combine magnetoencephalographic (MEG) measures of neural processing speed with magnetic resonance imaging (MRI) measures of white and grey matter in a large population-derived cohort to investigate the relationship between age-related structural differences and visual evoked field (VEF) and auditory evoked field (AEF) delay across two different tasks. Here we use a novel technique to show that VEFs exhibit a constant delay, whereas AEFs exhibit delay that accumulates over time. White-matter (WM) microstructure in the optic radiation partially mediates visual delay, suggesting increased transmission time, whereas grey matter (GM) in auditory cortex partially mediates auditory delay, suggesting less efficient local processing. Our results demonstrate that age has dissociable effects on neural processing speed, and that these effects relate to different types of brain atrophy.

[1] Medical Research Council, Cognition and Brain Sciences Unit, Cambridge CB2 7EF, UK. [2] Cambridge Centre for Ageing and Neuroscience, University of Cambridge and MRC Cognition and Brain Sciences Unit, Cambridge CB2 3EB, UK. [3] Department of Psychology, Harvard University, Harvard, Massachusetts 02138, USA. [4] Neuroscience Centre, University of Helsinki, Helsinki, FI-00014, Finland. [5] Division of Neuroscience and Experimental Psychology, School of Psychological Sciences, University of Manchester, Manchester M13 9PL, UK. Correspondence and requests for materials should be addressed to D.P. (email: darren.price@mrc-cbu.cam.ac.uk).
[†] A full list of consortium members appears at the end of the paper.

Age-related declines in cognitive abilities like fluid intelligence and working memory are well-documented, and a major burden for both older individuals and the societies they inhabit[1]. One potential common cause for these cognitive declines is a general slowing of information processing speed[2]. This slowing can be measured by behavioural responses in various tasks, which has been related to age-related atrophy of white matter (WM)[3–5] and grey matter (GM)[6–10]. Animal studies have proposed at least two mechanisms of age-related slowing: demyelination, which results in longer axonal transmission times between neurons[11,12], and changes in the neuron itself (such as increased hyperactivity), which results in reduced neural responsiveness[13]. These animal studies reinforce the importance of changes in both white and grey matter, and raise the possibility that these changes cause different types of neural slowing. Despite these links between WM, GM and behavioural slowing in humans, and between physiological changes and neural processing delays in animals, there is currently no evidence from studies of healthy ageing in humans that directly links differences in brain structure with differences in neural processing speed. Such evidence would provide important mechanistic insights into the causes of age-related slowing of information processing, and hence cognitive decline. We provide such evidence by combining magnetoencephalography (MEG) and magnetic resonance imaging (MRI) from a large sample of 617 population-derived healthy adults, distributed uniformly from 18 to 88 years of age, recruited from the Cambridge Centre for Ageing and Neuroscience (Cam-CAN; www.cam-can.org).

Age-related slowing of the neural response evoked by simple visual stimuli such as checkerboards has been observed using event-related potentials (ERPs) recorded with electroencephalography (EEG)[14–17]. Similar effects have also been found for auditory stimuli[18–21], complex visual stimuli such as faces[22–26] and olfactory stimuli[17,27]. Many studies have reported effects of age on early components of the ERP or event related field (ERF; for example, within 200 ms of stimulus onset)[15,17,28–30], while others have reported age effects on later components (typically 200–800 ms) without a corresponding increase in latency in the early components[18,23,25,26,31–34]. We propose that these reflect two distinct types of delay: constant and cumulative delay. Constant delay affects all time points equally, equivalent to a temporal shift of the whole evoked response (both early and late components). Cumulative delay, on the other hand, increases with post-stimulus time, and therefore is easier to detect for late than early components. Despite reports of cumulative delay in the literature[18,24], there has been no systematic comparison of constant and cumulative delay, and it is possible that they have different neuronal causes. Research on senescent monkeys suggests that age-related delay of the visual evoked field (VEF) has a cortical origin[13], while a review comparing electroretinogram, and cortical evoked potentials suggest that delays in humans originate in the retinogeniculostriate pathway[28]. Similarly, the causes of delay of the auditory evoked response may include contributions from peripheral, central auditory system or cortical functional deficits[35–39].

The study of age-related delays in ERPs/ERFs is further complicated by inconsistent findings in the degree of age-related delay of both early and late components (for review, see refs 17,25,26,28). There are several likely sources of this inconsistency. First, differences in information processing demands across experimental tasks (such as attending to faces versus ignoring auditory oddballs) mean that evoked responses are difficult to compare across studies. While clearly providing important information in their respective domains, complex tasks like face recognition are likely to recruit multiple cortical systems,

and it is unclear how each system contributes to the spatially integrated signal recorded by EEG/MEG. Second, age-related delays may differ according to the type of stimulus, for example, visual versus auditory stimuli, because age could potentially have differential effects on brain regions specialised for different sensory modalities. Even when results are consistent across different tasks and modalities, they are rarely compared directly within the same group of participants. Third, previous studies have tended to compare small groups of young versus old volunteers, rather than examine continuous differences across the adult lifespan, and tended to use volunteers who are self-selecting, rather than being representative of the population.

Yet another important source of inconsistency across studies is the method of measuring delay. One common measure is the latency of the peak (maximum) of an evoked component. This measure is very sensitive to noise however (being based on a small number of time points), so other techniques pool over several time points[23], for example, using the fractional area latency[40], or the slope or intercept of functions fit to parts of the evoked response. Here we use all (peristimulus) time points to estimate delay, improving robustness and sensitivity, and enabling estimation of second-order delay characteristics like constant and cumulative delay, which are not readily available from single peak latencies, and difficult to quantify when there are multiple, potentially overlapping temporal components.

To address some of the other limitations in the literature, we directly compare the effects of age across two types of task (Passive and Active) and on two types of stimuli (visual and auditory). Furthermore, we apply our novel method for simultaneously estimating constant and cumulative delay to a larger and more representative (opt-out) sample than is typically tested, spanning the whole adult lifespan. Finally, we relate these estimates of neural delay, for the first time, to structural estimates of both GM and WM on the same individuals.

More precisely, we use simple stimuli (visual checkerboards and pure tones) that are likely to activate only a few brain regions, and compare two tasks that differ in attentional demand: (i) a Passive viewing/listening session in which visual or auditory stimuli are presented separately, and do not require a response, versus (ii) an Active task in which the same stimuli are presented simultaneously, and require a motor response. We also use MEG, which has the same temporal resolution as the EEG, but has the advantage of higher spatial resolution, because magnetic fields are less spatially distorted by biological tissue than electrical fields[41], thereby increasing our ability to separate brain sources (see also ref. 26). Brain structural measures come from three types of MRI contrast: T1-weighted, T2-weighted and diffusion-weighted. The T1 and T2 data are combined to optimize estimation of local GM volume. Diffusion data are optimized for estimation of the mean kurtosis (MK) of the tissue's water diffusion[42,43], which is believed to offer a sensitive metric of age-related changes of WM microstructure, such as changes of cell membranes, organelles and the ratio of intra and extra-cellular water compartments[44]. Moreover, in contrast to standard diffusion tensor measures, diffusion kurtosis measures are robust to regions with a high concentration of crossing fibres[43,45].

Our main experimental hypotheses were that age is positively correlated with constant and/or cumulative evoked response delay in auditory and visual conditions, and that this relationship between age and delay is mediated by differences in WM microstructure or GM volume.

Here we use a novel technique to show that VEFs exhibit a constant delay, whereas AEFs exhibit a cumulative delay. Furthermore, WM microstructure in the optic radiation (connecting thalamus to the visual cortex) partially mediates age-related constant delay of the VEF, whereas grey matter

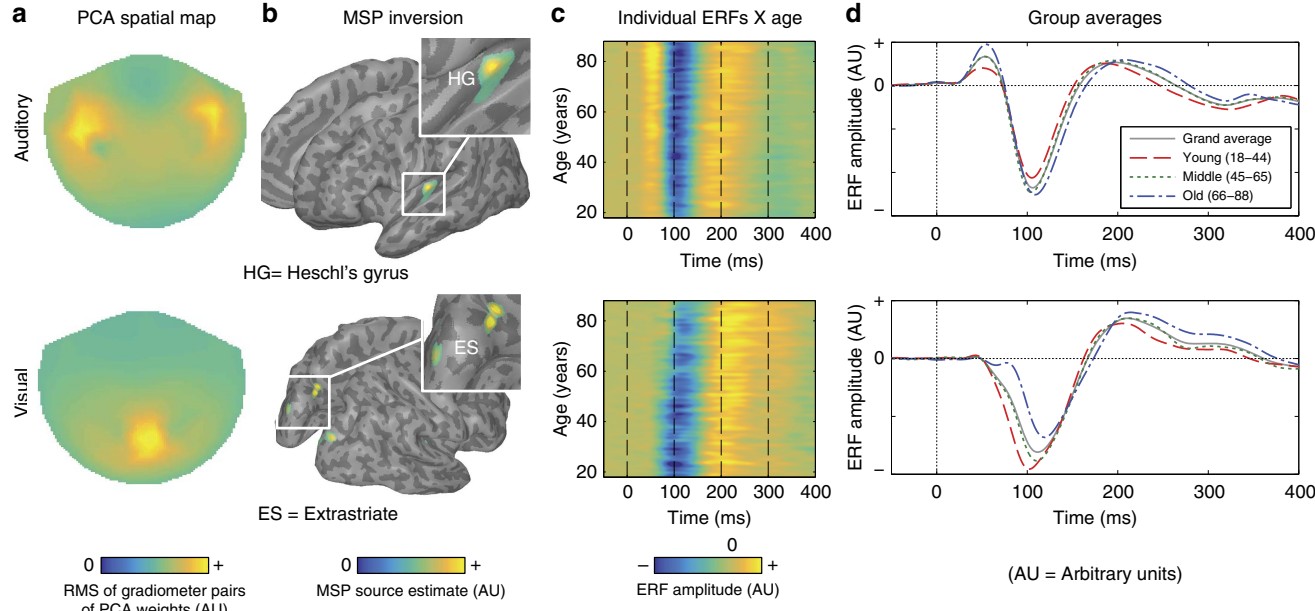

**Figure 1 | Principal component analysis (PCA) and delay estimation of auditory and visual evoked responses.** (**a**) Two-dimensional topographical MEG sensor plot of the first spatial component derived using PCA in the Passive task. Values represent the root mean square (RMS) of each pair of gradiometers. (**b**) Group Multiple Sparse Prior (MSP) source reconstruction based on the spatial component shown in **a** (cluser peak MNI coordinates: right HG = [ + 38, − 22, + 8], left = [ − 38, − 26, + 8]; right V2 = [ + 14, − 96, + 20], left = [ − 14, − 96, + 20]; ES right = [ + 16, − 74, + 24], left = [ − 16, − 74, + 20]). (**c**) Heat-maps illustrating the mean time course for each participant from the first temporal component of the PCA. Data are smoothed in the *y* direction for visualization only (Gaussian width = 5 subjects). (**d**) Group average time courses for each of three age groups (18–44, 45-65, 66–88 years). Because all of our analyses are based on principal components, the *y* axis of plots have arbitrary units.

structure in the superior temporal gyrus (STG; an auditory processing region) partially mediates cumulative delay of the AEF. These findings demonstrate a dissociation between the types of age-related delay in auditory and visual modalities. Furthermore, we show that these age-related delays are mediated by different aspects of age-related differences in brain structure: WM microstructure in the optic radiation partially mediates age-related visual constant delay, while grey-matter volume in the STG mediates age-related auditory cumulative delay.

## Results

**PCA-derived event related fields**. We start with data from the Passive task, in which auditory tones and visual checkerboards were presented in separate trials. To reduce our data set to a small set of meaningful components and improve the signal-to-noise ratio, principal component analysis (PCA) was performed on the trial-averaged event-related fields (ERFs) for each stimulus-type, after concatenating the data in the time dimension (Methods section). The first principal component explained 48% of the variance for auditory stimuli and 28% of the variance for visual stimuli, and entailed a spatial component for all participants (Fig. 1a) plus a separate time-course for each participant (Fig. 1c). The individual time courses were then averaged to create a template ERF for later model fitting (Fig. 1d). We used multiple sparse priors[46] to localize the spatial component, which revealed peaks in bilateral primary auditory cortex for the auditory stimuli and in bilateral extrastriate cortex for the visual stimuli (Fig. 1a,b). Importantly, the ERF-images in Fig. 1c show how the ERFs are delayed as age increases, with indication of a cumulative shift for auditory stimuli and a constant shift for visual stimuli; a difference that we formally quantify below.

To estimate constant and cumulative delay for each participant, a template fitting procedure was employed, in which the group average signal (grey line in Fig. 1d) was fit to each participant's

ERF by a combination of temporal displacement (constant delay) and temporal linear dilation (cumulative delay). Using a local gradient ascent algorithm, these two parameters were adjusted until the linear model fit ($R^2$) was maximized. Using $R^2$ as the utility function simplifies the fitting procedure, and allows simultaneous estimation of the amplitude offset and amplitude scaling (Methods section).

Constant and cumulative delay estimates for the AEFs and VEFs were correlated with age, using robust regression after removing outlying values for each modality separately (Fig. 2). In the visual condition, there was a significant effect of age on constant delay (percentage variance explained ($R^2$) = 0.11, $P < 0.001$, $N = 526$), but no effect of age on cumulative delay ($R^2 = 0.00$, $P = 0.996$, $N = 526$). In the auditory condition, on the other hand, there was a significant effect of age on cumulative delay ($R^2 = 0.15$, $P < 0.001$, $N = 577$), but not on constant delay ($R^2 = 0.00$, $P = 0.159$, $N = 577$). There was also a smaller but statistically significant increase in amplitude scaling of the AEF with age ($R^2 = 0.04$, $P < 0.001$, $N = 569$), a small increase in auditory amplitude offset ($R^2 = 0.02$, $P = 0.002$, $N = 562$), and an increase in visual amplitude offset ($R^2 = 0.09$, $P < 0.001$, $N = 491$; see Fig. 3). For conversions of these results to ms/year, and comparison with traditional peak latency estimates, see Supplementary Table 2.

We also repeated the PCA and ERF fitting steps for the Active task, in which the auditory and visual stimuli were presented simultaneously, and to which the participant responded with a key-press. This PCA produced similar results, though the simultaneous presentation of visual and auditory stimuli meant that the first spatial component reflected a mixture of responses to both stimulus-types (Supplementary Fig. 1). To better separate the auditory and visual responses in the Active task, and aid comparison with the Passive data, we applied the spatial weights of the PCA from the Passive data to estimate the time courses for each stimulus-type in the Active task. Importantly, age had a

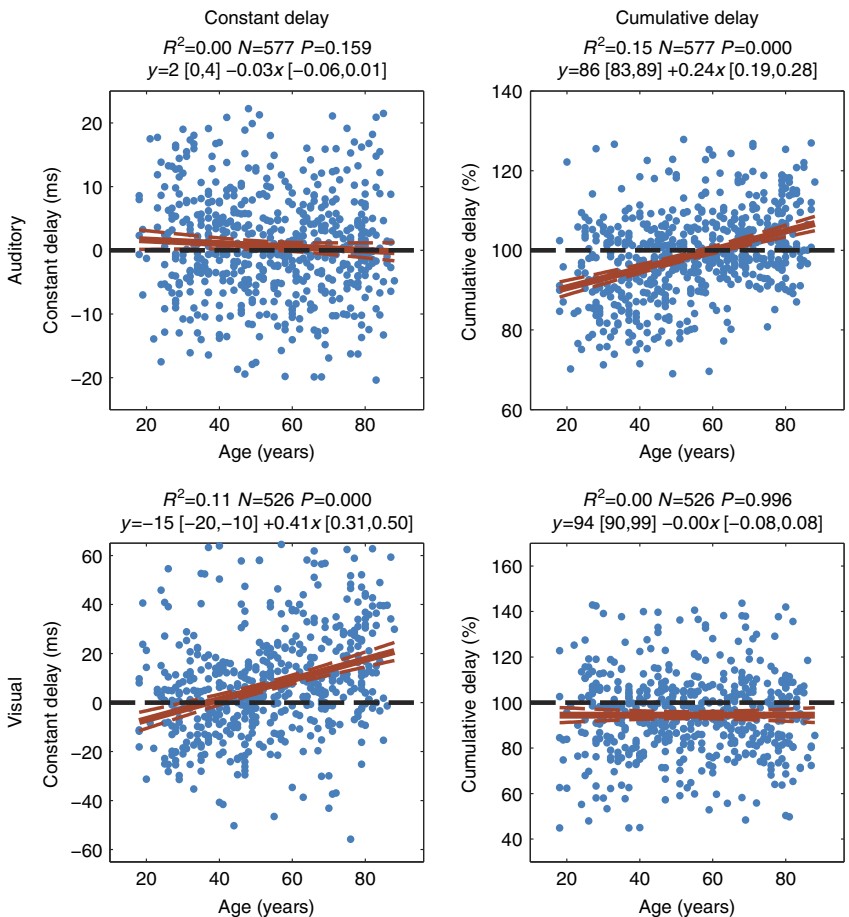

**Figure 2 | Robust regressions with age of the two delay parameters for each stimulus modality in the Passive task.** There is a significant effect of age on constant but not cumulative delay in the VEF, and a significant effect of age on cumulative but not constant delay in the AEF.

significant effect on the visual constant delay ($R^2 = 0.03$, $P < 0.001$, $N = 473$), but not visual cumulative delay ($R^2 = 0.00$, $P = 0.959$, $N = 473$), and on the auditory cumulative delay ($R^2 = 0.09$, $P < 0.001$, $N = 513$), but not auditory constant delay ($R^2 = 0.00$, $P < 0.188$, $N = 513$), replicating the pattern of significant age effects in the Passive task.

We also tested whether the task moderated the size of the age effect on the latency parameters. The effect of age on the auditory cumulative parameter was greater in the Active task than the Passive task, although no other delay parameters showed such an interaction between task and age (see Table 1). Nonetheless, for subsequent analyses below, we focus on the Passive data, where the auditory and visual responses are more easily separated by PCA.

**Testing assumptions of the delay model.** The template fitting method described here is conditional on several assumptions about the generators of the ERF. First, in applying PCA to our entire data set, we assume spatial stationarity of the evoked responses across the age range. It is possible that age-related changes in structural morphology result in changes in spatial distribution of the evoked fields. We address this issue in Supplementary Fig. 2b, where we compare the spatial distribution of signal variance between young and old participants, and did not observe any evidence of spatial non-stationarity. Furthermore, we did not find any qualitatively different results of ERF fitting when comparing templates derived from just young or just old participants (Supplementary Fig. 3).

Second, we assume temporal stationarity of all delay parameters. That is to say, the equations derived during fitting apply to the entire time-series (epoch), rather than distinct time windows of the evoked response. This rests on the hypothesis that evoked responses in MEG/EEG are the result of sustained, dynamic interactions between neuronal sub-populations within and across brain regions[47,48], producing damped oscillations, rather than being distinct and transient evoked components at the peaks and troughs of ERF/ERP waveforms. If this assumption is true, then changes to the physical characteristics of those neuronal populations (such as GM or WM integrity) are likely to lead to temporally extended delay characteristics. To test this hypothesis, we performed extensive simulations (Supplementary Fig. 4) that show when temporal stationarity holds, our template method is much more sensitive (in the presence of noise) than more traditional peak-based, peak-to-peak or fractional area measures of latency. We also confirmed, using traditional peak-based analyses, that constant and cumulative delay were evident in the experimental data, although peak-based measures are suboptimal for distinguishing between these two types of delay (Supplementary Fig. 5). Finally, we investigated whether there was evidence in our data against temporal stationarity. To do this, we repeated our template fitting approach on a shorter time window from 0 to 140 ms that only covered the first visual peak and the first two auditory peaks (excluding the later, more dispersed components). The results show that largely the same pattern of age effects on delay is observed in this early window as it is in the entire epoch (see Supplementary Fig. 2a), supporting the present assumptions.

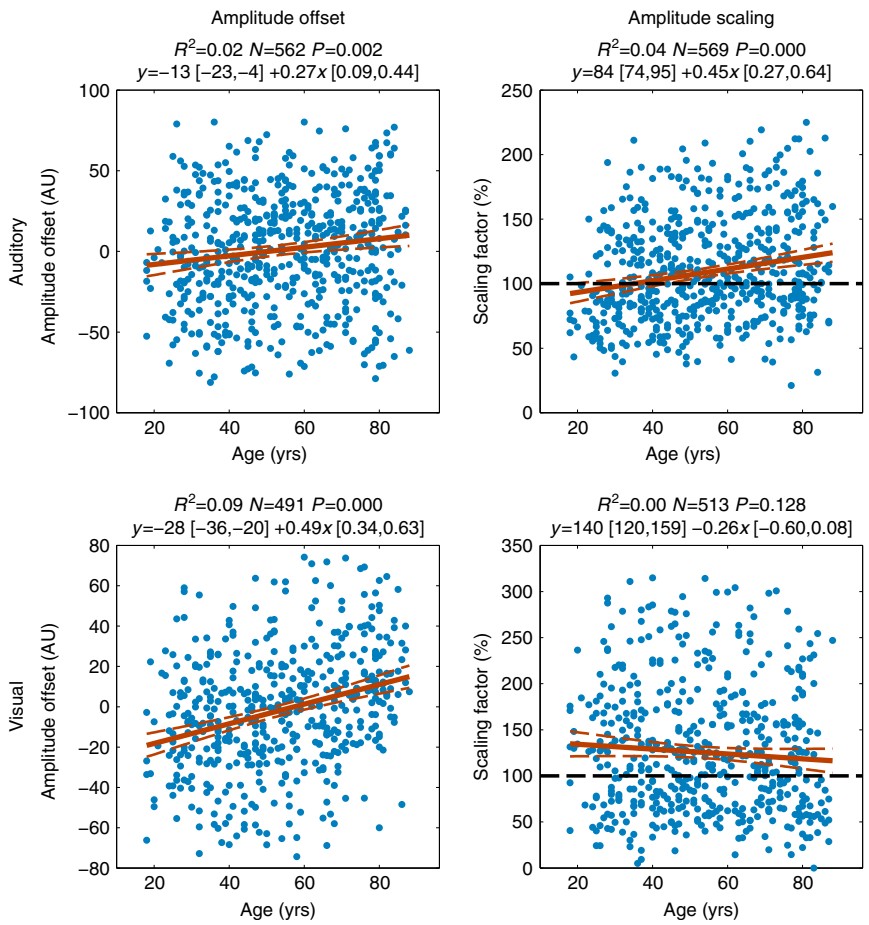

**Figure 3 | Amplitude offset and amplitude scaling parameters extracted during template fitting.** Amplitude offset was calculated by taking the mean difference between template and individual ERFs, while amplitude scaling indicates the amount the template needs to be scaled to minimize residual error between template and individual ERF.

**Table 1 | Comparison of ERF fitting results between Passive and Active tasks.**

|  | Slope: Passive − Active $r_s$ | Slope: Active $\beta$ [CI] ($R^2$) | Slope: Passive $\beta$ [CI] ($R^2$) | N |
|---|---|---|---|---|
| Auditory constant | 0.02 | − 0.03 (− 0.07, 0.01) (0) | − 0.03 (− 0.07, 0.01) (0) | 513 |
| Auditory cumulative | − 0.13** | 0.30 (0.22, 0.37) (0.09***) | 0.22 (0.17, 0.27) (0.14***) | 513 |
| Visual constant | 0.00 | 0.31 (0.1, 0.47) (0.03***) | 0.38 (0.28, 0.48) (0.10***) | 473 |
| Visual cumulative | 0.02 | 0.00 (− 0.11, 0.12) (0) | 0.00 (− 0.09, 0.08) (0) | 473 |

Statistical tests to compare the template fitting results from the active and passive sessions. The fitting parameters from the Active task were those obtained after applying Passive weights to the Active task data (Supplementary Fig. 1). To test whether the slope of the relationship between each delay parameter and age differed across the two tasks, we calculated the difference between Passive and Active tasks for each participant, and used a Spearman's correlation to test whether these difference scores were related to age. The age-effect was only significantly different across tasks for the auditory cumulative delay parameter (with a higher age-effect in the Active task). Data were removed from both Passive and Active data sets if an outlier existed in any column for a given modality, so that Passive and Active data sets contained equal numbers of participants. *$P<0.05$, **$P<0.01$, ***$P<0.001$

**Adjusting for confounding factors**. It is possible that the above effects of age on neural delays are simply a consequence of the known age-related changes in sensory acuity (despite our screening criteria, and use of lenses to correct vision; Methods section). We therefore correlated the neural delay estimates against separate, standardized measures of auditory and visual thresholds (Methods section). Results are summarized in Table 2. There was a negative relationship between visual acuity and visual constant delay (Spearman's correlation ($r_s$) = − 0.14, $P = 0.002$, $N = 524$), though this effect disappeared after adjusting for age ($r_s = 0.00$, $P = 0.993$, $N = 524$). Similarly, auditory acuity was negatively related to auditory cumulative delay ($r_s = − 0.21$, $P < 0.001$, $N = 575$), but not after adjusting for age

($r_s = − 0.05$, $P = 0.209$, $N = 575$). Importantly, however, both visual and auditory age-related neural delays remained significant after controlling for visual ($r_s = 0.30$, $P < 0.001$, $N = 524$) and auditory ($r_s = 0.31$, $P < 0.001$, $N = 575$) sensory acuity, respectively.

To check that the effects of age on neural delay estimates were not biased by effects of age on the estimated response amplitude (offset or scaling) or the estimated fit quality, we also repeated the partial correlation of neural delays with age after adjusting for these estimates. For both visual and auditory measures, the effects of amplitude scaling on delay were uncorrelated after controlling for age (auditory cumulative $r_s = − 0.02$, $P = 0.686$, 577; visual constant $r_s = − 0.01$, $P = 0.796$, $N = 526$) and the effects of age on

**Table 2 | Tests for the effect of potential confounds on delay.**

|                     | Age versus delay | Age versus acuity | Delay versus acuity | Delay versus acuity Cov = age | Age versus delay Cov = acuity | N |
|---------------------|:---:|:---:|:---:|:---:|:---:|:---:|
| Auditory cumulative | 0.37*** | −0.46*** | −0.21*** | −0.05 | 0.31*** | 575 |
| Visual constant     | 0.32*** | 0.43*** | −0.14** | −0.00 | 0.30*** | 524 |
|                     |  | Age versus amp | Delay versus amp | Delay versus amp Cov = age | Age versus delay Cov = amp |  |
| Auditory cumulative | - | 0.20*** | 0.06 | −0.02 | 0.37*** | 577 |
| Visual constant     | - | 0.07* | −0.03 | −0.01 | 0.32*** | 526 |
|                     |  | Age versus offset | Delay versus offset | Delay versus offset Cov = age | Age versus delay Cov = offset |  |
| Auditory cumulative |  | 0.13** | −0.16*** | −0.23*** | 0.41*** | 577 |
| Visual constant     |  | 0.30*** | 0.09 | −0.01 | 0.31*** | 526 |
|                     |  | Age versus RMSE | Delay versus RMSE | Delay versus RMSE Cov = age | Age versus delay Cov = RMSE |  |
| Auditory cumulative |  | 0.07 | 0.06 | 0.03 | 0.37*** | 577 |
| Visual constant     |  | −0.13** | −0.09* | −0.05 | 0.33*** | 526 |

Partial rank correlations (Spearman's Rho) demonstrating that the age versus delay relationships observed in the main results section are not accounted for by visual or auditory acuity, age-related changes in amplitude scaling (Amp), amplitude offset (Offset) or root mean square error (RMSE) of fit. Column 2 shows the relationship between age and the potential confounding variable (for example, auditory/visual acuity). Column 3 shows the relationship between evoked response delay and the potential confounding variable. Column 4 shows the relationship between age and delay, while controlling for age. Column 5 shows the relationship between age and delay while controlling for the confounding variable. In all cases, except for amplitude offset, controlling age abolishes the relationship between delay and the confounding variable. In all cases, controlling for the confounding variable has very little effect on the age–delay relationship. Auditory acuity measures were not available for 2 participants, who were excluded from the above analysis. * $P < 0.05$, **$P < 0.01$, ***$P < 0.001$.

delay remained significant after controlling for amplitude (auditory cumulative $r_s = 0.37$, $P < 0.001$, $N = 577$; visual constant $r_s = 0.32$, $P < 0.001$, $N = 526$). The same was true for amplitude offset and fit error (Table 2). Thus, there is no evidence that our age-related neural delays were caused by age-related differences in sensory acuity, response amplitude or offset, or goodness of template fit.

In summary, it appears that age has a qualitatively different effect on neural delay in auditory and visual modalities. Indeed, the auditory cumulative delay was only weakly correlated with visual constant delay ($R^2 = 0.02$, $P = 0.005$, $N = 491$), and this effect vanished after controlling for age ($R^2 = 0.00$, $P = 0.947$, $N = 491$), suggesting these two delay parameters are not intrinsically related, and are therefore likely to have separate underlying causes. To investigate possible causes, we turned to the MRI data on each participant.

**Mediation of neural latency by structural brain measures.** To test the hypothesis that brain structural changes account for some of the shared variance between age and neural delay, whole-head voxel-wise robust mediation analyses were performed[49]. Mediation analysis tests whether the relation—path c—between a predictor variable (X, age) and an outcome variable (Y, ERF delay) is significantly attenuated when the relation between X and a mediator variable (M, WM microstructure or GM volume)—path a—and the relation between M and Y—path b—are added to the model. Four separate models were tested at each voxel: one for each type of delay (auditory cumulative or visual constant) as the outcome and one for each brain measure (white or grey matter) as the mediator. All models included total intracranial volume (TIV) as a covariate of no interest. Mediation effect sizes were computed for every voxel, and a voxel-wise false detection rate (FDR) of 5% applied to correct for multiple comparisons. This threshold was further Bonferroni-corrected for multiple comparisons across the four models. Finally, voxels were also required to (i) show significant relations between age and mediator (path a) and between mediator and outcome (path b) at the same level of FDR correction, and (ii) fall within GM or WM masks.

The mediation effects of WM microstructure on the relationship between age and visual constant delay are displayed in Fig. 4a. Two clusters of significant voxels were found in the left retrolenticular part of the internal capsule (RIC; label 1), and the left posterior thalamic radiation (PTR; label 2). These paths together form the optic radiation projecting from the lateral geniculate nucleus (LGN) to the primary visual cortex (V1). The cluster centred on left RIC extended to left superior corona radiata and left superior longitudinal fasciculus, although the mediation effects here were generally lower. The cluster visible in left PTR also extended to splenium of corpus callosum, connecting left and right occipital cortices. There was no evidence that WM mediated the effects of age on auditory cumulative delay.

Mediation effects of GM on the relationship between age and auditory cumulative delay are displayed in Fig. 4b. One cluster was found comprising the left posterior STG (label 1), extending to middle temporal gyrus (MTG). Another cluster was observed in superior lateral occipital cortex (label 2), although effect sizes here were lower (peak = 12%) than in STG (peak = 26%). There was no evidence that GM mediated the effects of age on visual constant delay.

## Discussion

Using a novel analysis technique on MEG data from a large lifespan cohort, we discovered two distinct types of age-related neuronal delays in sensory-evoked responses: a constant delay in the visual evoked magnetic field (VEF) to checkerboards and a cumulative delay in the auditory evoked magnetic field (AEF) to tones. These delays occurred regardless of whether participants were passively encountering auditory and visual stimuli separately, or actively responding to them when both were presented concurrently, demonstrating that these age effects occur under variable levels of attention. After controlling for common age effects, these two types of delay were uncorrelated across individuals, suggesting dissociable causes. In support of this interpretation, we found that WM microstructure (MK, from diffusion-weighted MRI), primarily in the optic radiation (LGN to V1), mediated the effect of age on the visual constant delay, whereas GM volume (as estimated from T1- and T2-

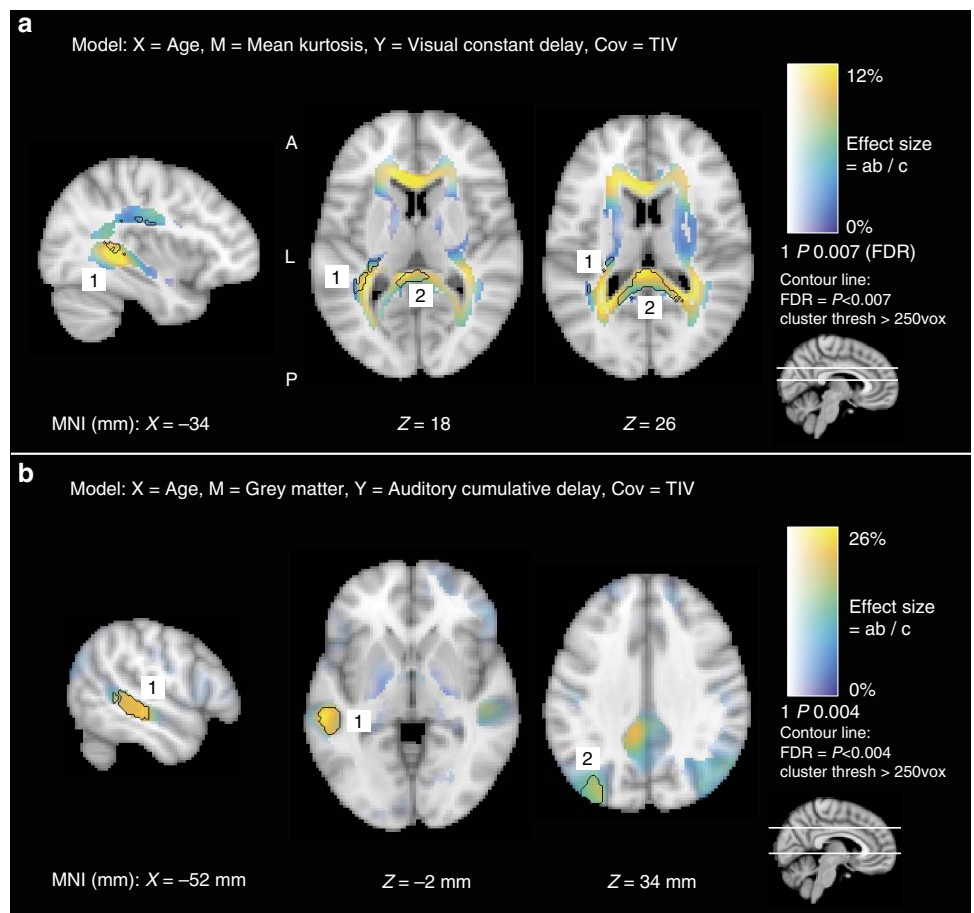

**Figure 4 | Whole-brain voxel-wise mediation analysis results.** Voxel hue corresponds to the effect size (% mediation effect), while opacity corresponds to the univariate *P* value. Clusters of at least 250 voxels that survive FDR correction are marked with a black border. (**a**) WM structure (MK) in the optic radiation (connecting LGN to V1) [label 1], and in splenium of corpus callosum [label 2], mediates the age versus visual constant delay relationship. (**b**) GM volume in the left posterior STG mediates the age versus auditory cumulative delay relationship [label 1]. Another cluster was observed in the left superior lateral occipital cortex, a region not involved in auditory processing [label 2]. All effects indicate positive mediation (that is, the age versus delay relationship is attenuated when including the mediator).

weighted MRIs), primarily in the posterior STG, mediated the effect of age on the auditory cumulative delay. We discuss these findings in light of prior behavioural, neuroimaging and animal studies.

The age-related neural slowing in the VEF and its relationship to WM microstructure in the optic radiation point to a delay in the arrival time of information communicated from the LGN to V1. Several aspects of our results support this conclusion. First, the main generator of the VEF was located in extrastriate visual cortex. Second, the slowing was characterized by a constant delay of the entire ERF, so that both early ($\sim$50–150 ms) and late ($\sim$150–400 ms) components of the evoked response were affected, consistent with a delayed arrival time. Third, the whole-brain voxel-wise mediation analysis revealed that microstructural differences in the left RIC and PTR partially account for the age-related delay. The RIC in particular is a major junction of fibres that transmit information from the LGN to visual cortex (and our use of diffusion kurtosis likely increased our sensitivity to regions with such crossing or fanning fibres). Furthermore, the lack of cumulative delay suggests that, once the information reaches visual cortex, it is processed at a normal rate (cf. auditory results, discussed below).

These results contrast with findings in the senescent macaque monkey[13], which revealed no WM atrophy, and no delay in the LGN-V1 pathway. Instead, spike-timing delays were related to

increased neuronal excitability and an increase in transmission time from V1 to V2. One reason for this discrepancy may be differences in the neural measures (for example, spiking rates versus the local field potentials recorded with MEG); another may be that humans have different neurobiological ageing profiles to macaques, highlighting the importance of investigating ageing *in vivo* in humans. Our results do not preclude the possibility that other information-carrying fibres and thalamic nuclei play a role in visual processing delays, but clearly point to age-related differences in WM microstructure being at least partially responsible for delays in information processing associated with healthy human ageing. Further, our findings support a review comparing pattern-evoked electroretinogram and cortical evoked potentials that concluded age-related visual deficits are the result of disruption to the retinogeniculostriate pathway[28].

The age-related cumulative delay observed in the auditory evoked response, along with the lack of constant delay, points to a different mechanism than that associated with the visual response. We suggest that this cumulative delay reflects a deficit in local processing within auditory cortex, specifically recurrent interactions between primary auditory cortex and higher order auditory regions. We again base this claim on several aspects of our analyses. First, the main source of the AEF was the primary auditory cortex, in-line with evidence that this region responds to pure tones[50]. This delay was mediated by GM volume in a higher

order auditory region, namely the STG. We also observed a cluster in left superior lateral occipital cortex, but since this region is not involved in processing of auditory stimuli, we suggest that collinear decline between this region and STG is responsible for this result, rather than any true functional link. The STG is too distant from the primary source to be considered the generator of the AEF, but is certainly anatomically connected to primary auditory cortex[51], suggesting that interactions with STG are responsible for the temporal dilation of the AEF. Indeed, the auditory system is organized into a functional hierarchy, beginning with processing of simple sounds (tone and frequency) in primary auditory cortex, before further processing in surrounding belt regions in the superior temporal lobe[47,50,52]. It is difficult to say with certainty, but the cumulative delay observed in the present study may therefore be linked to a disruption of neural function (possibly neural recovery times[53]) and dynamic interactions between primary auditory cortex and higher processing regions, resulting in delay that worsens over the duration of the evoked response, but does not affect the arrival time of information from the peripheral auditory system. Finally, we cannot completely rule out the contribution of central auditory processing deficits on our results[19,54]. However, central deficits are linked with age-related amplitude increases of the AEF, while our study demonstrated an independence of amplitude and latency. Further research is clearly needed to determine the contribution of central auditory processing deficits in cumulative delay of the AEF.

This idea of delays in local cortical interactions may also explain the results presented in recent studies[22,23] that have reported age-related cumulative delay for information accrual during processing of simple vs. complex visual stimuli after $\sim 90$–120 ms. These findings cannot be explained by the visual constant delay found here, because one would expect simple and complex stimuli to be equally delayed, leaving no net delay when contrasting them. We suggest that increasing stimulus complexity results in a greater dependence on recurrent communication between brain regions responsible for face discrimination. The speed at which this occurs, and therefore the temporal profile of the evoked response, would then be dependent on the integrity of those neural networks. Future studies directly comparing the delay profiles of responses across simple and complex discrimination tasks may help to shed light on these effects. Furthermore, computational modelling would help to understand the different neural mechanisms that could result in constant versus cumulative delay in different sensory systems.

One potential additional cause of age-related delay in evoked responses is the well-known age-related change in sensory acuity. If sensory acuity were an adequate predictor of neural delay, we would expect a relationship between acuity and delay to remain even after adjusting for age. However, while auditory acuity was weakly correlated with auditory cumulative delay, this correlation disappeared when accounting for age. The same was true for the relationship between visual acuity and visual constant delay. Most importantly, the correlations between age and both visual and auditory delays remained significant after adjusting for sensory acuity. These results suggest that sensory acuity does not play a significant role in our findings, which is consistent with previous studies arguing that optical and retinal factors cannot fully account for age-related delays in the visual evoked response[14–16,22,28].

We also investigated the effects of task on ERF fitting results, and found evidence that the effect of age on some latency estimates depends on the task in which latency is estimated: the slope of the relationship between auditory cumulative delay and age was higher in the Active task than Passive task. We cannot tell from our two paradigms whether this effect of task reflects

whether or not a response to the visual/auditory stimuli is required, and/or whether or not the visual and auditory stimuli are simultaneous. This is direct evidence for the concern raised in the Introduction that divergent results in the literature may reflect the use of quite different tasks. Nonetheless, the dissociable effects of age on visual constant versus auditory cumulative delay held across both of the present tasks (the slope for auditory cumulative delay was simply greater in the Active task), suggesting that the basic age effects on visual and auditory delay are somewhat invariant to the presence of the motor component of the task, or the concurrent presentation of auditory and visual stimuli.

The new method we used here to estimate delay made several assumptions. First, the use of PCA to decompose the data into a set of individual time courses and group spatial components assumes that the spatial components are stationary across the age range. We showed that this assumption holds in our data by demonstrating a high spatial correspondence between evoked response variance maps of young and old age groups. Second, the fitting of deriving constant and cumulative delay assumes temporal stationarity of delay parameters over the entire evoked response. To test this assumption, we fit ERFs from short time windows (0–140 ms) and found the same pattern of visual-constant and auditory cumulative delay. Furthermore, we employed a more traditional peak fitting method with both simulated and real data to show that, when treating peaks of the evoked response as separate components, constant and cumulative delay can still be derived, albeit with less reliability. Therefore, we conclude that the assumptions of temporal and spatial stationarity were met in our data, and that our method was suitable to derive independent measures of constant and cumulative delay.

Nonetheless, when interpreting these results, some caveats should be noted. First, while our findings go beyond previous studies in testing for age-related delays across both visual and auditory modalities and across two different tasks, our findings are nonetheless restricted to simple visual and auditory stimuli, which may not generalize to more complex stimuli that require more extensive neural processing. This might explain the age-related cumulative delay found previously in the ERP to faces[23], which likely involves a greater degree of recurrent processing between multiple visual cortical regions responsible for face perception.

Second, caution is warranted over interpretation of our mediation analysis. Mediation analysis is a statistical approach that cannot properly determine causality in the same way that an intervention might (for example, to lesion parts of the optic radiation and test effects on visual constant delay). Furthermore, mediation results from cross-sectional studies cannot be interpreted solely in terms of the ageing process[55], and have alternative explanations such as cohort effects. Whatever the precise role of age, our findings nonetheless demonstrate that there are at least two types of neural processing delay, which are unrelated across individuals, and have different relationships with white and grey matter in the brain structures associated with that processing.

Third, this study presents a novel investigation of the relationship between delays in human evoked responses and brain structure (as measured by MRI). As such, we had no specific hypotheses about the relative roles of grey versus WM, or different brain regions. Thus, even though our whole-brain search revealed statistically significant and mechanistically interpretable results, our findings should be regarded as exploratory. Future studies could take a more confirmatory approach to testing the role of specific structural properties in sensory processing delays.

Finally, note that we are not claiming that our present white and grey matter findings are a complete account of age-related slowing. Our voxel mediation effect sizes explained at most 26% of the age-related variance in response delays. It is possible that a larger proportion of variance in neural delay could be explained by combining brain measures across voxels, but even then, full mediation would be surprising since several other factors not measured here, such as altered neurotransmitter concentrations[56] and central auditory processing deficits[54] are also potential contributory factors. Nonetheless, our findings represent an important step forward in demonstrating dissociable types of age-related neuronal slowing, and generating mechanistic hypotheses for their causes.

To conclude, the present work fills a gap in the literature: evidence that in healthy humans, age-related delay of the electrophysiological response to stimulation is due to structural differences of functionally-relevant brain regions responsible for the transfer and processing of information. We have taken this a step further to show that neural delay should not be thought of as a unitary concept that affects all brain regions equally. Instead, ageing appears to be associated with regionally specific changes in characteristic neural responses, which are likely due to heterogeneous age-related changes across anatomical structures.

## Methods

**Participants.** Participants were recruited from a healthy population-derived sample from the Cam-CAN study (www.cam-can.org; see Shafto et al.[57] for a comprehensive explanation of the study design and experimental protocol). Ethical approval for the study was obtained from the Cambridgeshire Research ethics committee. Prior to the home interview, individuals give written informed consent for the study. Written informed consent is also given by participants at each scanning session. Participants were excluded based on several criteria: Mini Mental State Examination <25; failing to hear a 35 dB 1 kH tone in either ear; poor English language skills (non-native or non-bilingual speakers); self-reported substance abuse and serious health conditions (for example major psychiatric conditions, or a history of stroke or heart conditions); or MRI or MEG contraindications (for example, ferromagnetic metallic implants, pacemakers or recent surgery). Participants that did not take part in both the MEG and MRI sessions were also excluded. The final sample of N = 617 had an age range of 18–88 years at the time of first contact. For a *post hoc* analysis of power (number of participants needed to detect the present effect sizes), see Supplementary Fig. 6. Participants completed a Siemens HearCheck auditory screener consisting of three sound pressure levels (75dB, 55dB, and 35dB) at two frequencies (1000Hz and 3000Hz) without hearing correction. The Snellen sight test is also performed with corrected vision. In addition to screening tests at home interview stage, participants took both a visual (Snellen sight test, with vision correction) and auditory acuity test immediately preceding the MEG scan (see Shafto et al.[57] p7 for details). The scores from the home interview stage were also used in later statistical analysis of age-related neural delay to control for the possible confounds of age-related differences of visual/auditory acuity (2 participants with missing acuity data were removed; see Table 2).

**Audio-visual tasks.** The visual stimulus consisted of two circular checkerboards presented simultaneously to the left and right of a central fixation cross (34 ms duration × 60 presentations). The diameter of each circular checkerboard subtended an angle of 3°, their centroids were separated by 6° on the horizontal plane, and the checks had a spatial frequency modulation of 2 cycles per degree. Visual stimuli were presented using a Panasonic PT-D7700 DLP projector (1,024 × 768, 60 Hz refresh rate) outside of the magnetically shielded room (MSR), projected though a waveguide onto a back-projection screen placed 129cm from the participant's head. The stimulus onset was adjusted for the 34 ms (2 refreshes at 60 Hz) delay induced by the projector. The auditory stimulus was a binaural tone (300 ms duration; 20 presentations of 300, 600 and 1,200 Hz; 60 total presentations), with a rise and fall time of 26 ms, and presented at 75 dB sound pressure level (measured using an artificial ear). The stimulus onset was adjusted for the 13 ms delay for the sound to reach the ears. The first session involved the Active task, in which participants were presented with both types of stimuli concurrently, and asked to respond by pressing a button with their right index finger after stimulus onset. The reaction times (RTs) are analysed in Supplementary Table 1, though note that the Active task was not explicitly speeded. The stimulus-onset asynchrony varied randomly between 2 and 26 s, to match an fMRI version of same task (see Taylor et al.[58] for more details). The session lasted 8 min and 40 s in total.

The second session was the Passive task. Participants experienced separate trials with either the visual or auditory stimulus, which they were asked to passively observe (no response required). Visual and auditory trials were pseudo-randomly ordered, with a stimulus onset asynchrony (SOA) that varied randomly between 0.8 and 2 s. Given the simultaneous presentation, evoked responses were adjusted by the average delay of 23 ms (13 ms auditory and 34 ms visual). The session lasted ~2 min.

**PCA based latency analysis.** The 2D (time × sensor) matrices for each participant were concatenated along the time dimension. PCA was performed with columns as signals and rows as observations to produce a set of time domain signals for each trial-type. The *n*th principal component was then reshaped to give individual trial averages for each participant. The principal component weights represent the degree to which each channel contributes towards the *n*th principal component. Given that the relationship is linear, the weights can also be used for source localization. Since the simple sensory-evoked sources are likely to be distributed across multiple regions, we used multiple sparse priors, implemented in SPM12, which is capable of recovering multiple sparsely distributed generators of ERFs (ref. 46). Each participant's MRI was coregistered to their MEG data using three anatomical fiducial points (nasion, and left and right pre-auricular points) that were digitized for the MEG data and identified manually on the MRIs. Lead fields were calculated using a single-shell model based on the deformed spheres approach[59].

**ERF Fitting.** The aim of the fitting procedure was to obtain estimates for two types of delay: constant delay, defined as delay that affects all time points equally (modelled using a 0th order delay parameter); and cumulative delay, defined as delay that is linearly dependent on the time point at which it is measured (modelled using a 1st order delay parameter). Note that the delay terms could be of any order, but limiting the delay estimates to 0th and 1st order terms reduces the danger of over-fitting. ERF fitting was performed using in-house code written in MATLAB. First, a template ERF was computed from the data as the trial-averaged ERF for a given principal component, averaged across all participants. Because this template represents the group average, approximately equal numbers of participants will have negative and positive relative delay parameters when fitting their individual ERFs. A template fitting algorithm was designed to iteratively alter the temporal characteristics of the template signal $\mathbf{s}(t)$ by:

$$\hat{\mathbf{s}}(t) = \mathbf{s}\left(t_0 - \tau_{CON} + \frac{[t - t_0]}{\tau_{CUM}}\right). \quad (1)$$

where $\tau_{CON}$ is the constant delay, $\tau_{CUM}$ is the cumulative delay and $t_0$ is a stationary point in time for any value of $\tau_{CUM}$, and was constant across all tests (see below). Cubic spline interpolation was used to obtain $\hat{\mathbf{s}}$ for any given set of delay parameters.

For each individual time-course $\mathbf{s}(t)$, the parameters were adjusted iteratively using a local maximum gradient ascent algorithm until the convergence criteria was satisfied. Starting parameters were chosen to correspond to the null hypothesis that no delay would be observed compared to the group average ($\tau_{CON} = 0; \tau_{CUM} = 1$), and delay parameters were adjusted by a predefined quantity ($\tau_{CON} = \pm 20$ ms; $\tau_{CUM} = \pm 10$ %), giving 4 fits per iteration. A value of $t_0 = 50$ ms was fixed on the basis of the typical latency for information to reach sensory cortices (for example, P1m). Fit was determined using linear regression, and the parameter values that gave the best $R^2$ fit were chosen as starting points in the next iteration. Using regression simplified the fitting procedure, since amplitude scaling and mean were determined from the regression model's beta estimate and intercept, respectively. If none of the parameters resulted in a better fit than the current best fit, then the magnitude of the parameter adjustments were reduced by a factor of 0.75 and the process was repeated. Convergence was achieved when the $R^2$ fit improvement over the current best fit was <1e-6. This method helped to reduce the occurrence of spurious outliers, because the gradient ascent was constrained to converge into the maxima closest to the starting point. Results were visually inspected to ensure optimal fitting was achieved. Outlier parameter values were handled during statistical testing, described below.

**Robust regression.** For each delay parameter, outliers were identified based on the boxplot rule (± 1.5 times the inter-quartile range) and removed from the analysis. An outlier in either the constant or cumulative delay estimate resulted in the participant being rejected from the analysis for a given experimental condition (visual or auditory). For all correlation analyses, robust methods were employed to control for remaining extreme values. Robust regression was implemented using MATLAB Statistical Toolbox (LinearModel.fit). A bi-square weighting function with a tuning constant of 4.685 was used to weight cases based on their residual error from an ordinary least squares fit, then repeated until convergence.

For the purpose of comparing our results with those in the literature using more conventional measures, the linear equation describing the age-related change in constant and cumulative delay can be converted back to peak latency, $l$, by combining the age versus constant/cumulative delay regression equations (Results section) with Equation 1 (where $t$ becomes the peak-of-interest, $a$ is age, and $\tau_{CON}$ and $\tau_{CUM}$ are replaced with the linear equations obtained from our main analysis):

$$l = t_0 + \beta_{0\,CON} + a\beta_{1\,CON} + \beta_{0\,CUM}(t - t_0) + a\beta_{1CUM}(t - t_0). \quad (2)$$

Separating the constant and age dependent terms gives the coefficients of the linear equation $l = \beta_0 + a\beta_1$:

$$\beta_0 = t_0 + \beta_{0\,CON} + \beta_{0\,CUM}(t - t_0) \quad (3)$$

$$\beta_1 = \beta_{1CUM}(t - t_0) + \beta_{1CON} \quad (4)$$

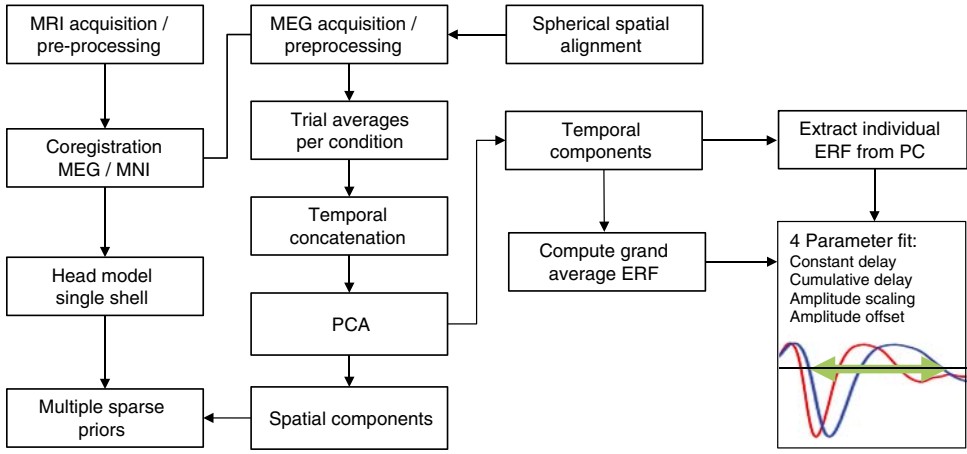

**Figure 5 |** Flow chart diagram illustrating the processing steps involved in the analysis of MEG data.

where $t$ is the peak-of-interest (for example, for P2m, $t = 200$—obtained from the ERF template), and $\beta_{0\,CON}$, $\beta_{0\,CUM}$, $\beta_{1\,CON}$ and $\beta_{0\,CUM}$ are the intercepts and slope betas from the age versus delay regression equations. Confidence intervals can also be converted using the same formula.

**MRI Data.** MRI data were acquired using a Siemens 3T TIM TRIO (Siemens, Erlangen, Germany) with a 32-channel head coil at the MRC Cognition & Brain Sciences Unit (CBU), Cambridge, UK. Anatomical images were acquired with a resolution of 1 mm³ isotropic using a T1-weighted MPRAGE sequence (TR: 2250 ms; TE: 2.98 ms; TI: 900 ms; 190 Hz; flip angle: 9°; FOV: 256 × 240 × 192 mm; GRAPPA acceleration factor: 2), and a 1 mm³ isotropic T2-weighted SPACE sequence (TR: 2800 ms; TE: 408 ms; flip angle: 9°; FOV: 256 × 256 × 192 mm; GRAPPA acceleration factor: 2). Diffusion-weighted images (DWIs) were acquired with a twice-refocused spin-echo sequence, with 30 diffusion gradient directions for each of two $b$-values: 1,000 and 2,000 s mm$^{-2}$, plus three images acquired with a $b$-value of 0. These parameters are optimized for estimation of the diffusion kurtosis tensor and associated scalar metrics. Other DWI parameters were: TR = 9,100 milliseconds, TE = 104 milliseconds, voxel size = 2 mm isotropic, FOV = 192 mm × 192 mm, 66 axial slices, number of averages = 1.

All MRI data were analysed using the SPM12 software (www.fil.ion.ucl.ac.uk/spm), implemented in the AA 4.2 batching software (https://github.com/rhodricusack/automaticanalysis).

The T1 and T2 images were initially coregistered to the MNI template using a rigid-body transformation, and then combined to segment the brain into 6 tissue classes: GM, WM, cerebrospinal fluid, bone, soft tissue and residual noise. The GM images were then submitted to diffeomorphic registration (DARTEL) to create group template images, which was then affine-transformed to the MNI template. To accommodate changes in volume from these transformations, the GM images were modulated by the Jacobean of the deformations to produce estimates in MNI space of the original local GM volume.

The DWI data were first coregistered with the T1 image and then skull-stripped using the BET utility in FSL (http://fsl.fmrib.ox.ac.uk/fsl/fslwiki/). Linear fitting of a higher order tensor was then used to estimate MK (using in-house code). Images of the diffusion metrics were then normalized to MNI space using the DARTEL + affine transformations from the T1 + T2 pipeline above.

**MEG scanning and pre-processing.** Data were collected continuously using a whole-head Elekta Neuromag Vector View 306 channel MEG system (102 magnetometers and 204 planar gradiometers; Elekta, Neuromag, Helsinki, Finland), located in a MSR at the CBU. Data were sampled at 1kHz with a highpass filter of 0.03 Hz. Recordings were taken in the seated position. Head position within the MEG helmet was estimated continuously using four head-position indicator coils to allow for offline correction of head motion. Two pairs of bipolar electrodes were used to record vertical and horizontal electrooculogram signals to monitor blinks and eye-movements, and one pair of bipolar electrodes to record the electrocardiogram signal to monitor pulse-related artefacts. Instructions and visual stimuli were projected onto a screen through an aperture in the front wall of the MSR. Participants were given MEG-compatible glasses to correct their vision. Auditory stimuli were presented binaurally via etymotic tubes. Motor responses were made via a custom-built button box with fibre optic leads. For a schematic diagram of the MEG processing pipeline, see Fig. 5.

Temporal signal space separation (tSSS; MaxFilter 2.2, Elekta Neuromag Oy, Helsinki, Finland) was applied to the continuous MEG data to remove noise from external sources and from head-position indicator coils (correlation threshold 0.98, 10-s sliding window), for continuous head-motion correction (in 200-ms time windows), and to virtually transform data to a common head position ('-trans

default' option with origin adjusted to the optimal device origin, [0, + 13, − 6]). MaxFilter was also used to remove mains-frequency noise (50 Hz notch filter) and to automatically detect and virtually reconstruct any noisy channels. Data were then imported into Matlab using SPM12. We identified physiological artefacts from blinks, eye-movements and cardiac pulse using the logistic Infomax independent components analysis (ICA) algorithm implemented in EEGLAB. This was done by identifying those ICs whose time courses and spatial topographies correlated highly with reference time courses (correlation greater than three s.d.'s from mean) and spatial topographies (correlation greater than two s.d.'s from mean), respectively, for each of the above artefact types (run via in-house code, detect_ICA_artefacts.m).

Data were then filtered using a two-pass 5th order Butterworth filter (1-32 Hz) implemented in fieldtrip, epoched (time window: − 100 to 500 ms peristimulus time), and the average baseline (− 100 to 0 ms) was subtracted from the data. Trial averaged responses of evoked amplitude were computed for each channel, participant and condition of the experiment. This resulted in a two-dimensional matrix for each participant (channels × time). All subsequent analyses were carried out using data from the 204 gradiometer channels, since these are more sensitive to superficial sources than the magnetometer sensors, and the sensory cortices of interest here are superficial. ERF fitting on both the Passive and Active task data was performed on the principal components of the channel level data (see subsection on PCA based latency analysis, below).

**Whole-brain voxel-wise mediation analysis.** Images in MNI space of MK from the DWI pipeline, and of local GM volume (GMV) from the T1 and T2 pipeline, were smoothed (12 mm FWHM) and entered into a whole-brain voxel-wise robust mediation analysis implemented using the M3 Mediation Toolbox (http://wagerlab.colorado.edu)[49]. A three path model was used with age as the predictor variable (X), delay as the dependent variable (Y; outliers removed as above), and anatomical data as the mediator variable (M). Since TIV was correlated with age ($R^2 = 0.01$, $P = 0.03$), it was also included as a covariate to control for the possible confounding effects of head size on structural statistics.

Mass univariate robust mediation was computed per voxel using the robust fit option of the M3 toolbox (10,000 bootstrapped samples per voxel) to generate path data (paths: a = X−>M, b = M−>Y, c' = X−>Y, c = ab + c'), and associated $P$ values calculated per voxel. Four models were tested, all with age as the predictor (X) and TIV as covariate of no interest, which correlated with age ($R^2 = 0.01$, $P = 0.03$, $N = 617$). The mediator variable (M) was voxel data from either the white or GM images. WM volumes were masked using the JHU ROIs (all ROIs were included); GM volumes were masked using the Harvard Oxford GM atlas. Only those voxels falling within the mask were entered into the mediation model. The FDR threshold for each image was calculated from the resulting $P$ value maps using the M3 mediation toolbox. Statistical maps were thresholded according to these $P$ value thresholds, and clusters with fewer than 250 suprathreshold voxels were excluded. Mediation effect sizes were calculated using the formula, $M_{effect} = 100ab/c$, which represents the mediation effect on the c path of including $M$ in the model (resulting in c') as a percentage of the total direct effect (c). A value of 100% indicates full mediation.

**Data availability.** Data from the Cam-CAN project is available from the managed-access portal at http://camcan-archive.mrc-cbu.cam.ac.uk, subject to conditions (see website). Analysis scripts are available in Supplementary Information accompanying this publication. For a complete description of Cam-CAN data and pipelines, see Taylor et al.[58]

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

## Acknowledgements

The Cambridge Centre for Ageing and Neuroscience (Cam-CAN) research was supported by the Biotechnology and Biological Sciences Research Council (BB/H008217/1); R.N.H. is additionally supported by the UK Medical Research Council (MC_A060_5PR10); DP is supported by the UK Medical Research Council (MC_A060_5PR13). We are grateful to the Cam-CAN respondents and their primary care teams in Cambridge for their participation in this study. We also thank colleagues at the MRC Cognition and Brain Sciences Unit MEG and MRI facilities for their assistance. Finally, we thank Dr Rousselet and two other reviewers for their helpful comments.

## Author contributions

Cam-CAN conceived the study. Cam-CAN piloted the study and collected the data. J.R.T., N.W., D.P. and R.H. collated and preprocessed the data. D.P. wrote the in-house MEG analysis software and analysed the data. D.P., R.N.H., L.K.T., N.H., K.C., N.W., M.T. and J.R.T. wrote the paper.

## Additional information

**Competing interests:** The authors declare no competing financial interests.

## Cam-CAN consortium

Carol Brayne[2], Edward T. Bullmore[2], Andrew C. Calder[2], Rhodri Cusack[2], Tim Dalgleish[2], John Duncan[2], Fiona E. Matthews[2], William D. Marslen-Wilson[2], James B. Rowe[2], Meredith A. Shafto[2], Teresa Cheung[2], Simon Davis[2], Linda Geerligs[2], Rogier Kievit[2], Anna McCarrey[2], Abdur Mustafa[2], David Samu[2], Kamen A. Tsvetanov[2], Janna van Belle[2], Lauren Bates[2], Tina Emery[2], Sharon Erzinglioglu[2], Andrew Gadie[2], Sofia Gerbase[2], Stanimira Georgieva[2], Claire Hanley[2], Beth Parkin[2], David Troy[2], Tibor Auer[2], Marta Correia[2], Lu Gao[2], Emma Green[2], Jodie Allen[2], Gillian Amery[2], Liana Amunts[2], Anne Barcroft[2], Amanda Castle[2], Cheryl Dias[2], Jonathan Dowrick[2], Melissa Fair[2], Hayley Fisher[2], Anna Goulding[2], Adarsh Grewal[2], Geoff Hale[2], Andrew Hilton[2], Frances Johnson[2], Patricia Johnston[2], Thea Kavanagh-Williamson[2], Magdalena Kwasniewska[2], Alison McMinn[2], Kim Norman[2], Jessica Penrose[2], Fiona Roby[2], Diane Rowland[2], John Sargeant[2], Maggie Squire[2], Beth Stevens[2], Aldabra Stoddart[2], Cheryl Stone[2], Tracy Thompson[2], Ozlem Yazlik[2], Dan Barnes[2], Marie Dixon[2], Jaya Hillman[2], Joanne Mitchell[2] & Laura Villis[2]

