## [Peer Review File · Nature Communications]

Reviewers' comments:

Reviewer #1 (Remarks to the Author):

This is one of the most exciting papers on cognitive ageing I have ever read. It tackles the fundamental processing speed theory of ageing, its system specificity, and how it relates to structural brain changes. However, unlike all studies I have ever found on the topic, processing speed is not measured using behavioural tasks: instead, system specific estimations are provided using MEG in two modalities. The sample size is very large, uniform and representative. So essentially the article could be published as it is: the main question is one of the most important in cognitive ageing, the results are very compelling and very well analysed, and the paper is overall very well written and documented. This is the sort of papers I'd like to read more often! Also, thanks to the authors for including all the figures in the text, including the supplementary ones - this makes reviewing much easier and should be mandatory in every journal.

Below I list some comments and a few recommendations. But I'll start with one main point that requires clarification:

In figure 1, in addition to an increase in onset latency, the early negative visual response (blue band) seems to shorten in duration with increasing age. If this is the case, this effect should be associated with a negative temporal linear dilation in older age, and thus a negative slope in the age-related regression. Unless the second positive component shows the opposite pattern, but this is more difficult to assess from figure 1. Can you clarify exactly how the ERFs are changing with age? To help answer this question, the illustration of the effects in panel C might benefit from a different colormap. For instance, have you tried the new default Parula colormap with that figure?

<<http://blogs.mathworks.com/steve/2014/10/20/a-new-colormap-for-matlab-part-2-troubles-with-rainbows/>> I'm not sure but it might help visualise the extent of the second positive component, which at the moment is difficult to read against the light greenish background.

Also, to complement panel 1C, in panel D it would be more informative to show averages for different age groups, to get a better sense of how the evoked responses differ with age. It is possible that the age-related differences in the visual domain are a bit more complex than suggested by the current analysis. Maybe I'm completely wrong, in which case better illustrations would help make a stronger case.

Finally, can you clarify the differences between the results in figure 1 and supplementary figure 1? The apparent shortening of the first negative visual component in Figure 1C seems absent in the active task (supplementary figure 1a, 2nd column, first row), unless the PCA weights from the passive take were applied (supplementary figure 1b), in which case the blue band is not only progressively phase shifted as one goes up along the y axis, but also shrinks progressively. That's just me eyeballing the data but the differences seem substantial enough.

Introduction

This section is very clear and brings together all the crucial dichotomies. The distinction between constant and cumulative delays is extremely useful, and one that seems promising to look back at existing datasets.

If you have room, you could complement the listing of age-related ERP delays with this nice example from the olfactory domain:

Murphy, C., Morgan, C.D., Geisler, M.W., Wetter, S., Covington, J.W., Madowitz, M.D., Nordin, S. & Polich, J.M. (2000) Olfactory event-related potentials and aging: normative data. *Int J Psychophysiol*, 36, 133-145.

Thanks for citing our work. To clarify, the conclusion of our studies (your references 21-23) is that there are large age-related differences within 200 ms, starting roughly around 120 ms post-stimulus onset. Recently, we confirmed the lack of evidence for age-related differences in visual ERP onsets to a face vs. texture contrast, typically occurring circa 90 ms irrespective of age: Bieniek, M.M., Bennett, P.J., Sekuler, A.B. & Rousselet, G.A. (2015) A robust and representative lower bound on object processing speed in humans. *The European journal of neuroscience*.

This discrepancy with your constant delay finding in the visual domain suggests the possibility of a dissociation between the timing of the evoked response to one stimulus, and the time it takes for the brain to dissociate between two categories. The introduction is probably not the right place to bring up this point; your discussion addresses the potential distinction between simple and complex stimuli, and given the novelty of your results I'm not even sure there is much you could be speculating about...

To support the point "Despite several reports of cumulative delay in the literature", I think the most appropriate reference is 23 - see in particular figure 10
<<http://journal.frontiersin.org/article/10.3389/fpsyg.2010.00019/full>>
Same comment for the discussion.

Results

Figure 2: I would suggest to report the equation of the linear fit in each panel, with confidence intervals for the slope and intercept estimates, and to remove the p values, which do not provide much information. Personally, I care about effect sizes.

Comparisons between tasks: did you compare the intercept and slope estimates between tasks? This can be done using a percentile bootstrap approach - see e.g. Rand Wilcox's robust estimation 2012 book. I'm happy to help with the code if you have any question.

Figure 3: it would be more informative to show non-thresholded maps. See recommendations and Matlab code here e.g.:

Allen, E.A., Erhardt, E.B. & Calhoun, V.D. (2012) Data visualization in the neurosciences: overcoming the curse of dimensionality. *Neuron*, 74, 603-608.

Discussion

Overall I find the discussion well balanced, with a clear flag for most limitations.

Although you measured visual acuity, typically in the ageing literature on perception, contrast sensitivity and senile miosis are also considered. This could be flagged as well.

Reference 13: you mention the lack of association between speed and white matter that could mediate age-related differences in processing speed in macaque monkeys. You should also point out the lack of age-related differences in the latencies of early V1 responses (layer 4). In ref 13 they report age-related delays in V1 neurones from outside layer 4, suggesting that ageing affects intra-cortical communication, whereas speed is unaffected between the retina and V1. They also report increased delays in V1 compared to V2, suggestive of a cumulative profile. So there are more discrepancies between your work and previous findings in monkeys than what you currently suggest in the discussion. However, at this stage we have so little evidence to draw clear conclusions that I don't see a problem with these discrepancies.

Methods

I am not an MRI specialist, so I cannot comment on the structural analyses. For the rest:

Which ICA algorithm did you use?

What was the order of the Butterworth filter?

Although not a problem, given the nature of the analyses I don't see the point of reconstructing (interpolation?) noisy channels.

I understand given the scale of the project that drastic choices had to be made, but personally I would have avoided a passive task, especially as used as second task: passive means participants can revert to a default task of their choice, which is unknown to the experimenters, or perform the same task they used in session 1. I would have chosen detection vs. discrimination, to have a more controlled level of engagement.

Which Matlab function did you use for robust regression? Can you clarify why you removed univariate outliers before applying robust regression? Robust methods are designed to handle outliers, so removing them should be unnecessary, unless some values were physiologically implausible and thus not worth considering at all.

Can you confirm that the MEG data are available online? It's not very clear from the website associated with ref 46. It would also increase the impact of your research if you shared the analysis code online.

Related to the analysis code, and given the large number of participants, it would be useful to run data driven simulations to determine the minimum number of participants necessary to detect the effects reported here. Is that something you have considered? Would be worth adding it to the paper if you have time, or add a technical note somewhere else later.

The bootstrap is not in itself more robust to outliers: associated with robust estimators, it provides in many situations more accurate confidence intervals than classic parametric

methods, but this is not always the case.

writing / typos:

"ERP fitting" should be "ERF fitting"

I find "the fact that" particularly ugly, e.g.

"owing to the fact that magnetic fields are less susceptible than electrical fields to spatial distortion in biological tissue". This could be rephrased as "because magnetic fields are less spatially distorted by biological tissues than electrical fields".

I always sign my reviews: Guillaume Rousselet

Reviewer #2 (Remarks to the Author):

This is a well written and interesting manuscript that describes age-related increases in sensory evoked response latency and how that latency relates to measures of white and grey matter in a sample of young, middle-aged and older adults. This is an important topic and this study extends prior research using EEG and MEG by relating age-related changes in sensory evoked response latency with changes white and grey matter integrity. That being said, the current version of the manuscript does not provide sufficient methodological details.

Explaining how their new analysis technique better captures age-related changes in sensory evoked responses is needed. As I understand, their technique is used to estimate the delay in sensory evoked responses, and it takes into account the entire epoch used to capture the age-related delay in sensory evoked response. This is a major departure from prior work, which focuses primarily on peaks and troughs as well as inter-peak measurements (e.g., Woods et al. 1993). It remains unclear to what extent their approach is more appropriate than prior techniques that used peak latency. Comparing their new procedure to estimate age-related increased in response latency with more conventional measures would illustrate the value-added and/or the benefits of theirs vs. the usual, and would greatly strengthen the manuscript. It would also allow the reader to compare with prior research showing age-related increased in N1 and P2 waves for visual and/or auditory stimuli.

The new procedure used to obtain the ERP template and to calculate the latency should be presented earlier in the text. Moreover, it is unclear to what extent the new technique takes into account changes in ERP amplitude, which would alter the morphology and might be confounded with differences in latency. For instance, the auditory P1 wave notoriously increases in amplitude with age, but its peak shows little sensitivity to aging. Visual evoked responses show less age-related increased in amplitude. Could the age-related effects on constants and cumulative delays be partly accounted for by different patterns of age-related changes in ERP amplitude for visual and auditory stimuli? The manuscript should consider this in the discussion.

The age-related change in latency reported in this study appears small compared to those

reported for the auditory N1 and P2 wave. Again, it would be useful to show conventional measurement of the N1 and P2 wave. The peak-to-peak difference in latency can also be reported as it can be used as an index of cumulative delay.

The behavioral data from the attention tasks should be analyzed and reported in the text. It is important to demonstrate that the current sample shows the typical age-related increases in response time. It would also be interesting to explore the relation between changes in ERP, white and grey matter on response time.

Figure 1 shows the template used to extract latencies, but this template appears to be limited to sensory evoked responses. The manuscript does not present the ERPs elicited by target stimuli during the active listening task. These are particularly important because they should be closely related to behavior.

Listing "CamCAN" as an author is peculiar. I understand that the data are coming from a consortium and this is already indicated in the acknowledgement. Hence, there is no need to add the consortium as a co-author, is there? This would bring the definition of co-authorship to a new level of confusion, inconsistent with usual criteria for authorship of scientific work.

In the method section, the description of the auditory and visual tasks should precede the sections on MRI and MEG. Also, event-related potential (ERP) should be replaced with event-related field (ERF). Lastly, the reference to N1 and P2 should include the lower case letter m (e.g., N1m).

The description of the visual task should include the visual angle for stimulus presentation as well as the rise/fall time and intensity for the auditory stimuli. The inter-stimulus interval should be reported as well as the steps (e.g., stimuli were presented with ISI ranging from 500 ms and 1000 ms with 10 ms steps, rectangular distribution). The term passive task does not make much sense and should be replaced with passive recording condition (i.e., no response required).

Line 73, I would replace "Some" with "Many." Indeed, there are many studies reporting effects of age on sensory evoked response latency, including the N1 and P2 waves. It is correct to say that few studies have reported age effects on middle latency evoked responses and that age effects on N1 latency are more equivocal with some studies reporting age-related increased in latency while other do not.

Line 152-153, it is unclear whether the ERP traces are actually brain waves or the most dominant component from the PCA. Given the arbitrary unit, one would assume the later, but this needs to be clarified. Also, whenever possible avoid the use of arbitrary unit for y-axis.

Line 257, delete "Finally"

Line 401, please add a comma after e.g.

Line 418, replace "are" with "were"

Reference section: The references on frequency following responses (FFR) do not seem appropriate given that the present studies focuses on cortical evoked responses. The ref # 42 also seems out of place. I would highly recommend that the authors more carefully review prior work on aging using EEG, MEG and fMRI. Notably, there are several highly relevant MEG studies on sensory systems and aging that are currently omitted from their review of the literature.

Reviewer #3 (Remarks to the Author):

This is a very interesting research project, attacking from a valid angle, and delivering compelling evidence for its arguments. Its main strength is arguably the unusually large sample size of >500.

The authors show using a arguably more robust all-timepoints-based estimation technique that auditory and visual cortical-response changes slow down differentially in older age (cross-sectionally, of course, as the authors correctly concede); suggesting different aging mechanisms behind cumulative and constant response delay, respectively.

They link these cortical slowings in interesting, and still relatively simple (i.e., parsimonious) mediating-analysis models, to brain structural data and are thus able to link changes in certain brain areas.

I am very much in favour of this manuscript, and very much enjoyed reading it.

I offer comments below that the authors should find helpful in re-considering some aspects of their analysis and presentation. They mostly relate to aspects of model fitting and statistics.

-- A corollary from the cumulative-delay idea: Should the first indication of differential age-shift in audio vs visual not be nicely visible if the ERO in Fig 1d or elsewhere would show some measure of dispersion? I would expect that the dispersion (while increasing probably for both, aud and vis, with time) increases more drastically with time in audition.

Anyhow, a way to have Fig. 1d reflect the effect would be very welcome.

-- This bring me to one of my more major methodological concerns: Is the fitting method, as intriguing as it sounds, really fair? Could it be improved by using some form of cross-validation/train-and-test approach? As I understand it, the average ERP across all subjects was first derived and used as a template. Now, this average ignores the dispersion that (we know from the rest of the paper) is increasing with time in audition (but not in vision), because of age (for early time points, young and old listenres have the "same" ERP at a given time point, for later time points they increasingly diverge).

We could thus assume that later portions of the (particularly auditory) average ERP are a worse representation of the individual ERP, and this might or might not bias/influence parameter estimates and R^2 .

Maybe the authors have a good reply to this at hand, but maybe an approach where the template is not biased towards the hypothesis (worse fit later on, thus maybe favouring one

or the other model) might be preferable? A cross-validation approach, where the template is somehow derived independent of the to-be-fitted data, might help.

-- The one thing that this paper lacks in order to be of markedly higher relevance would be a plausible (computational) model of how the differential effects of age on auditory and visual responses can come about. This is asking for a lot, I know; but given the [senior] authors' experience in this field, I was expecting something like this. The section of the neurobiology behind these effects had to remain speculative, and I am not convinced why (i) auditory responses should behave so differently from visual responses; the most marked difference lies in the number of synapses and complexity of wiring *before* the afferent signal reaches cortex, but the authors seem to imply/observe that ...

(ii) age-related delay should only start in cortex. There are by now many studies linking or trying to link ageing to alterations in the sensorineural periphery, especially in the auditory domain (often or mostly employing brainstem responses, frequency-following response, e.g. from the Kraus lab, the Shinn-Cunningham lab, Terry Picton and others). Why should the neurobiological explanation offered here, e.g. altered feedback/inhibition between STS and HG, apply not earlier in the processing hierarchy?

-- I would like to see the authors acknowledge more upfront the (modest) effect size of their main findings. They do have the numbers to crunch out statistical effects. But this does not change the fact that an R^2 of .1 is only 10% of the variance explained. (I agree that this is noteworthy, but it is also clear that in more conventional sample sizes the same effect, an r of .3, would hardly pass conventional levels of significance and thus go often unnoticed in the literature.)

Related, they speak of "mediation effect sizes [per voxel]". Where are they reported/what are they used for? A p -value of course is *not* an effect size, but I am uncertain what the authors refer to here.

-- I am not convinced by using linear (robust) regression methods on variables that are clearly, by their range of values, not normal. (See e.g. Sensory Acuity, bound to $[-\infty; 1]$, or percentages like visual amplitude, bound to $[0; \infty]$). The authors should apply adequate transforms (log, logit, arcsine, etc.) or apply rank regression. This might change conclusions in particular in some voxels, where relations now thought to be meaningfully linear turn out not to be and vice versa.

-- It speaks in favour of the authors that they resisted the temptation to posthoc "spin" detailed hypotheses about which brain structures (white and/or grey matter) would potentially influence cortical slowing -- but the approach nevertheless leaves us with a "fishing expedition" (albeit with adequate statistical control of type I error), so the brain areas identified and discussed might be taken by some readers with a grain of salt.

-- very minor: Some figures, at least figure 1, contain a somewhat ugly mixture of typefaces, Arial and Calibri.

Reviewers' comments:

Reviewer #1 (Remarks to the Author):

This is one of the most exciting papers on cognitive ageing I have ever read. It tackles the fundamental processing speed theory of ageing, its system specificity, and how it relates to structural brain changes. However, unlike all studies I have ever found on the topic, processing speed is not measured using behavioural tasks: instead, system specific estimations are provided using MEG in two modalities. The sample size is very large, uniform and representative. So essentially the article could be published as it is: the main question is one of the most important in cognitive ageing, the results are very compelling and very well analysed, and the paper is overall very well written and documented. This is the sort of papers I'd like to read more often! Also, thanks to the authors for including all the figures in the text, including the supplementary ones - this makes reviewing much easier and should be mandatory in every journal.

We thank Dr Rousselet for his positive comments, and the time he devoted to our submission.

Below I list some comments and a few recommendations. But I'll start with one main point that requires clarification:

In figure 1, in addition to an increase in onset latency, the early negative visual response (blue band) seems to shorten in duration with increasing age. If this is the case, this effect should be associated with a negative temporal linear dilation in older age, and thus a negative slope in the age-related regression. Unless the second positive component shows the opposite pattern, but this is more difficult to assess from figure 1. Can you clarify exactly how the ERFs are changing with age? To help answer this question, the illustration of the effects in panel C might benefit from a different colormap. For instance, have you tried the new default Parula colormap with that figure? <<http://blogs.mathworks.com/steve/2014/10/20/a-new-colormap-for-matlab-part-2-troubles-with-rainbows/>> I'm not sure but it might help visualise the extent of the second positive component, which at the moment is difficult to read against the light greenish background.

Also, to complement panel 1C, in panel D it would be more informative to show averages for different age groups, to get a better sense of how the evoked responses differ with age. It is possible that the age-related differences in the visual domain are a bit more complex than suggested by the current analysis. Maybe I'm completely wrong, in which case better illustrations would help make a stronger case.

We have changed the colormaps on all figures to parula as suggested (see for example new Figure 1 below). This has improved the readability of the colormap plots, and also makes printing in black and white possible. We have also plotted three age groups in panel d. We thank Dr Rousselet for this suggestion.

We have also plotted ERFs from three age groups in Supplementary Figure 1 (also included below). The effect of age on the first negative visual response appears mainly as a shift (delay) and attenuation of amplitude, but it is possible there is a shortening too, which may reflect the influence of the later positive component. This raised the possibility that

differences in amplitude affect early and late components differentially (see also response to Reviewer 2), potentially confounding the pattern of constant and cumulative delay observed in the main results. We therefore repeated the fitting procedure with shorter time windows to exclude late components (0-140ms; marked on Supplementary Figure 1 ERF plots below). The same pattern of auditory cumulative delay and visual constant delay was observed, in line with findings in the main results section. There was a significant negative effect of age on visual cumulative delay, although the effect size was very small ($R^2 = 0.01$, $p=0.002$).

Supplementary Figure 1. a) ERFs (first temporal component of PCA) are plotted for three equally-sized age groups (18-43, 44-64, 65-88yrs). No age-related delay is apparent in the P1m of the auditory ERF, while the N1m and P2m show increasing delay of their respective peaks. Our template fitting procedure makes the assumption that cumulative delay affects both early and late components. However, it is possible that a violation of this assumption (e.g., delay only affecting P2m, but not P1m or N1m) would result in spurious effects of age on cumulative delay. To test this possibility, we estimated delay parameters based on a shorter time window of 0-140ms (marked) to exclude late components (e.g., P2m) from the analysis. For both auditory and visual ERFs, the same dominant pattern, of age-related auditory cumulative delay, and age-related visual constant delay, was observed. b) 2D sensor plots showing the signal variance across sensors for each of the three age groups, along with a difference plot showing the difference between young and old groups. The old group has higher signal variance around auditory sensors for auditory stimuli, but lower variance around visual sensors for visual stimuli.

However, there is negligible shift in the spatial distribution of variance, as indicated by the high correlation coefficients between young and old topographies.

Finally, can you clarify the differences between the results in figure 1 and supplementary figure 1? The apparent shortening of the first negative visual component in Figure 1C seems absent in the active task (supplementary figure 1a, 2nd column, first row), unless the PCA weights from the passive take were applied (supplementary figure 1b), in which case the blue band is not only progressively phase shifted as one goes up along the y axis, but also shrinks progressively. That's just me eyeballing the data but the differences seem substantial enough.

We have replotted the data using parula colormap in Supplementary Figure 2, also included below. This figure should make it clear that some mixing of signals occurs in the active task, owing to the difficulty of separating auditory and visual components when both stimuli are simultaneous. This is why we concluded that our method of using PCA to separate sources breaks down when stimuli are simultaneous, due to high temporal correlation in the visual and auditory ERFs. After separating components by applying the PCA weights from the passive task however, the auditory and visual colormaps of the active task, and associated timecourses, look much more similar to the passive session.

Supplementary Figure 2: Principal component topography and time-series plots to show that using PCA on the active task data results in mixing of signals, and that this problem is alleviated in the passive session. **a)** Principal components from the active task. Auditory and visual components are not easily separable due to correlations in the evoked response timecourses. **b-c)** Principal component topographies from the visual and auditory conditions of the passive session. Timecourses are shown for both the passive and active task after applying sensor weights from the passive session (see row labels).

With respect to the amplitude changes observed in visual ERFs with age (N1m reduced; P2m increased), this appears to be due to a combined age-related change in the mean amplitude offset and amplitude scaling. The fitted amplitude offset and scaling are now plotted in Supplementary Figure 6, since these parameters are not of primary interest for our questions about latency. This further illustrates the usefulness of the template fitting approach, in allowing the extraction of latency estimates independent of other effects of age on the evoked response.

Supplementary Figure 6. Amplitude offset and amplitude scaling parameters extracted during template fitting. Amplitude offset was calculated by taking the mean difference between template and individual ERFs, while amplitude scaling indicates the amount the template needs to be scaled to minimise residual error between template and individual ERF.

Introduction

This section is very clear and brings together all the crucial dichotomies. The distinction between constant and cumulative delays is extremely useful, and one that seems promising to look back at existing datasets.

If you have room, you could complement the listing of age-related ERP delays with this nice example from the olfactory domain:

Murphy, C., Morgan, C.D., Geisler, M.W., Wetter, S., Covington, J.W., Madowitz, M.D., Nordin, S. & Polich, J.M. (2000) Olfactory event-related potentials and aging: normative data. Int J Psychophysiol, 36, 133-145.

We thank the reviewer for bringing this paper to our attention, which we now cite.

Thanks for citing our work. To clarify, the conclusion of our studies (your references 21-23) is that there are large age-related differences within 200 ms, starting roughly around 120 ms post-stimulus onset. Recently, we confirmed the lack of evidence for age-related differences in visual ERP onsets to a face vs. texture contrast, typically occurring circa 90 ms irrespective of age: Bieniek, M.M., Bennett, P.J., Sekuler, A.B. & Rousselet, G.A. (2015) A robust and representative lower bound on object processing speed in humans. The European journal of neuroscience.

This discrepancy with your constant delay finding in the visual domain suggests the possibility of a dissociation between the timing of the evoked response to one stimulus, and the time it takes for the brain to dissociate between two categories. The introduction is probably not the right place to bring up this point; your discussion addresses the potential distinction between simple and complex stimuli, and given the novelty of your results I'm not even sure there is much you could be speculating about... To support the point "Despite several reports of cumulative delay in the literature", I think the most appropriate reference is 23 - see in particular figure 10 <<http://journal.frontiersin.org/article/10.3389/fpsyg.2010.00019/full>> Same comment for the discussion.

We have addressed this point further in the discussion.

Results

Figure 2: I would suggest to report the equation of the linear fit in each panel, with confidence intervals for the slope and intercept estimates, and to remove the p values, which do not provide much information. Personally, I care about effect sizes.

We have included the equation in each of the figures, as well as plotting the prediction bounds for each of the linear fits.

Comparisons between tasks: did you compare the intercept and slope estimates between tasks? This can be done using a percentile bootstrap approach - see e.g. Rand Wilcox's robust estimation 2012 book. I'm happy to help with the code if you have any question.

The intercept estimate does not provide any useful information when comparing between tasks if the templates are derived separately as it is a relative shift from the group mean (see Supplementary Figure 7). However, the slopes between tasks should, in theory, be more stable. Therefore we have included slope comparisons in Supplementary Table 3 (below). The slope of the age effect on auditory cumulative delay was significantly higher in the active than passive task.

	Slope: Passive – Active r_s	Slope: Active β [CI] (R^2)	Slope: Active β [CI] (R^2)	N
Aud. Const.	.02	-.03 [-.07, .01] (0)	-.03 [-.07, .01] (0)	513
Aud. Cumtv.	-.13**	.30 [.22, .37] (.09***)	.22 [.17, .27] (.14***)	513
Vis. Const.	.00	.31 [.15, .47] (.03***)	.38 [.28, .48] (.10***)	473
Vis. Cumtv.	.02	.00 [-.11, .12] (0)	.00 [-.09, .08] (0)	473

*P<0.05, **P<0.01, ***P<0.001

Supplementary Table 3. Statistical tests to compare the template fitting results from the active and passive sessions. The fitting parameters from the active task were those obtained after applying passive weights to the active task data (see Supplementary Figure 2). To test whether the slope of the relationship between each delay parameter and age differed across the two tasks, we calculated the difference between passive and active tasks for each participant, and used a Spearman's correlation to see if these difference scores were related to age. The age-effect was only significantly different across tasks for the auditory cumulative delay parameter (with a higher age-effect in the active task). Data were removed from both passive and active tasks if an outlier existed in any column for a given modality, so that passive and active datasets contained equal numbers of participants.

Therefore, we have some evidence that the effect of age on some latency estimates does depend on the task in which latency is estimated. We cannot tell from our two paradigms whether this effect of task reflects 1) whether or not a response to the visual/auditory stimuli is required and/or 2) whether or not the visual and auditory stimuli are simultaneous. We now report these further findings in the Results and Discussion. Nonetheless, the important point remains that the dissociable effects of age on visual constant delay but not visual cumulative delay, and one auditory cumulative delay but not auditory constant delay, are invariant to our two different tasks.

Figure 3: it would be more informative to show non-thresholded maps. See recommendations and Matlab code here e.g.: Allen, E.A., Erhardt, E.B. & Calhoun, V.D. (2012) Data visualization in the neurosciences: overcoming the curse of dimensionality. Neuron, 74, 603-608.

We have re-plotted our mediation results using the method you suggested. We have also taken the code from the above reference, and created a generic Matlab function. This will be released with publication of the manuscript, so others can benefit from it and hopefully further improve it.

Discussion

Overall I find the discussion well balanced, with a clear flag for most limitations.

Although you measured visual acuity, typically in the ageing literature on perception, contrast sensitivity and senile miosis are also considered. This could be flagged as well.

We added the following text to the discussion: Other possible contributing factors include contrast sensitivity and senile miosis (which affects retinal luminance). While these factors were not measured in the present study, Bieniek et al.²² found that age-related delays to faces are likely to be cortical in origin, and not caused by changes in pupil size or luminance, so we conclude that this is not a likely explanation for our results.

Reference 13: you mention the lack of association between speed and white matter that could mediate age-related differences in processing speed in macaque monkeys. You should also point out the lack of age-related differences in the latencies of early V1 responses (layer 4). In ref 13 they report age-related delays in V1 neurones from outside layer 4, suggesting that ageing affects intra-cortical communication, whereas speed is unaffected between the retina and V1. They also report increased delays in V1 compared to V2, suggestive of a cumulative profile. So they are more discrepancies

between your work and previous findings in monkeys than what you currently suggest in the discussion. However, at this stage we have so little evidence to draw clear conclusions that I don't see a problem with these discrepancies.

Our discussion of visual delay in the macaque now reads:

These results contrast with findings in the senescent macaque monkey¹³, which revealed no white-matter atrophy, and no delay in the LGN-V1 pathway. Instead, spike-timing delays were related to increased neuronal excitability and an increase in transmission time from V1 to V2. One reason for this discrepancy may be differences in the neural measures (e.g., spiking rates versus the local field potentials recorded with MEG); another may be that humans have different neurobiological ageing profiles to macaques, highlighting the importance of investigating ageing *in-vivo* in humans. Our results do not preclude the possibility that other information-carrying fibres and thalamic nuclei play a role in visual processing delays, but clearly point to age-related differences in white-matter microstructure being at least partially responsible for delays in information processing associated with healthy human ageing. Further, our findings support a review comparing pattern-evoked electroretinogram (ERG) and cortical evoked potentials that concluded age-related visual deficits are the result of disruption to the retinogeniculostriate pathway²⁸.

Methods

I am not an MRI specialist, so I cannot comment on the structural analyses. For the rest:

Which ICA algorithm did you use?

We used the logistic infomax ICA algorithm as implemented in EEGLAB, as now stated.

What was the order of the Butterworth filter?

We used a 5th-order butterworth filter, as implemented using Fieldtrip toolbox, as now stated.

Although not a problem, given the nature of the analyses I don't see the point of reconstructing (interpolation?) noisy channels.

This is a standard part of the MaxFilter program that implements Signal Space Separation (SSS). SSS uses a spherical harmonic basis-set, which is the proper interpolation method (given Maxwell's equations) for reconstructing magnetic signals (in bad channels) from nearby signals. Such reconstruction is necessary for the PCA we performed on concatenated data from all subjects, which requires complete datasets for every subject.

I understand given the scale of the project that drastic choices had to be made, but personally I would have avoided a passive task, especially as used as second task: passive means participants can revert to a default task of their choice, which is unknown to the experimenters, or perform the same task they used in session 1. I would have chosen detection vs. discrimination, to have a more controlled level of engagement.

We agree, but the task was based on multiple different constraints from the larger CamCAN project. We will consider this point in future.

Which Matlab function did you use for robust regression? Can you clarify why you

removed univariate outliers before applying robust regression? Robust methods are designed to handle outliers, so removing them should be unnecessary, unless some values were physiologically implausible and thus not worth considering at all.

Robust methods should indeed cope with outliers, but the degree to which they cope can still be influenced by those outliers, and we felt there were indeed implausible values that should be excluded. We used the LinearModel.fit algorithm in Matlab stats toolbox, with the robust option (bisquare weighting, tuning constant of 4.685 which is the default value). This has been clarified in the methods section.

Can you confirm that the MEG data are available online? It's not very clear from the website associated with ref 46. It would also increase the impact of your research if you shared the analysis code online.

We now include instructions on how to access the data via our online portal. The paragraph reads: "Data from the CamCAN project is available from the managed-access portal at <http://camcan-archive.mrc-cbu.cam.ac.uk>. Analysis scripts are available in supplementary materials accompanying this publication."

Related to the analysis code, and given the large number of participants, it would be useful to run data driven simulations to determine the minimum number of participants necessary to detect the effects reported here. Is that something you have considered? Would be worth adding it to the paper if you have time, or add a technical note somewhere else later.

We ran the simulations as suggested, and included these in Supp. Fig. 5:

The bootstrap is not in itself more robust to outliers: associated with robust estimators, it provides in many situations more accurate confidence intervals than classic parametric methods, but this is not always the case.

We have removed this statement from the methods section.

**## writing / typos:
"ERP fitting" should be "ERF fitting"**

This has been amended throughout

**I find "the fact that" particularly ugly, e.g.
"owing to the fact that magnetic fields are less susceptible than electrical fields to spatial distortion in biological tissue". This could be rephrased as "because magnetic fields are less spatially distorted by biological tissues than electrical fields".**

This has been amended as suggested

Reviewer #2 (Remarks to the Author):

This is a well written and interesting manuscript that describes age-related increases in sensory evoked response latency and how that latency relates to measures of white and grey matter in a sample of young, middle-aged and older adults. This is an important topic and this study extends prior research using EEG and MEG by relating age-related changes in sensory evoked response latency with changes white and grey matter integrity. That being said, the current version of the manuscript does not provide sufficient methodological details.

We thank the reviewer for their positive comments, and the time they devoted to our submission.

Explaining how their new analysis technique better captures age-related changes in sensory evoked responses is needed. As I understand, their technique is used to estimate the delay in sensory evoked responses, and it takes into account the entire epoch used to capture the age-related delay in sensory evoked response. This is a major departure from prior work, which focuses primarily on peaks and troughs as well as inter-peak measurements (e.g., Woods et al. 1993). It remains unclear to what extent their approach is more appropriate than prior techniques that used peak latency. Comparing their new procedure to estimate age-related increased in response latency with more conventional measures would illustrate the value-added and/or the benefits of theirs vs. the usual, and would greatly strengthen the manuscript. It would also allow the reader to compare with prior research showing age-related increased in N1 and P2 waves for visual and/or auditory stimuli.

We agree that it would be helpful to compare our template-fitting method to previous component-based measures. We have conducted several tests to compare the efficacy of template-fitting versus peak-fitting, in terms of both peak latencies and fractional area latencies, and included them as additional supplementary information.

Firstly, we show that, if delays do affect the whole epoch, as our template-fitting method assumes, then peak latencies alone are unable to distinguish constant from cumulative delays, while linear fitting of peak-to-peak estimates are much less sensitive in the presence of noise (Supplementary Figure 3).

However, it is also possible that the template-fitting assumption (that delays affect the whole epoch) is not true, and different components can be delayed independently. To see if this was the case in our data, we re-ran the peak-fitting procedures in Supplementary Figure 3 on the real data (Supplementary Figure 4 below).

Supplementary Figure 3: In order to compare the performance of the present template-fitting with previous peak-fitting approaches, we performed several tests on an idealised simulated evoked response (panel **a**). The simulated ERF was generated by adding together Gaussian peaks of varying parameters: amplitude (A), latency (L) and width (W) [peak 1: A = -1, L = 100ms, W = 25ms; peak 2: A = .5, L = 200ms, W = 35ms; peak 3: A = -1, L = 300ms, W = 100ms]. Peak fitting was achieved by defining three time windows (labelled in **a**) centred on each of the three peaks with a window of +/-50ms. To simulate our hypothesised age-effects, the ideal ERF was then resampled (using equation 1 from methods) with simulated constant delay (11 points ranging from -10 to +10ms) and cumulative delay (11 points ranging 0.95 to 1.05). For each combination of constant and cumulative delay, eight estimates of delay were displayed on 2D grids as follows: **b-c**) With template-fitting, the orthogonal gradients confirm that when assumptions about the delay types are met (i.e. that delay affects the whole epoch), and no noise is present in the data, constant and cumulative delay are perfectly separable. The values obtained are also accurate, so that the value in the grid at location [+10, 1.05] reads 10 in **b** and 1.05 in **c**. **d-f**) Peak latencies alone however are affected by both constant and cumulative delay, the influence of each depending on the time point under observation. **g**) Fractional area latency (FAL) was calculated by finding the point at which the integral of the absolute value of the

timecourse reached half of the maximum of the total integral. FAL was also unable to distinguish between constant and cumulative delay. **h-i**) By fitting all three peak latencies with a linear function, one can derive estimates of constant delay (intercept) and cumulative delay (slope, analogous to peak-to-peak latencies). This method is also accurate when no noise is present in the data, but performs worse than template-fitting in the presence of noise, as shown in **(j-k)**, where delay parameters were fixed (const. = +10ms, cumtv. = +1.05) and the fitting procedure repeated 10,000 times with 10 levels of white noise with a variance ranging from 0 to 5 times the variance of the simulated timecourse (1/SNR). The mean of each batch of 10,000 tests is plotted as a solid line, with standard deviations plotted as semi-transparent area around the mean. It is clear that template fitting provides more accurate estimates of constant and cumulative delay, even in the presence of relatively high levels of noise.

Supplementary Figure 4: The same method used in Supplementary Figure 3 was applied to the real data (though there were only two clear components in the visual case). **a**) Peak latencies for auditory and visual components. The effect of age on auditory peak delays is larger for later than earlier peaks, consistent with a cumulative delay. Visual latencies reveal some age-related delay in peak 1 and 2: Although R^2 is lower in peak 2 ($R^2=0.04$), the slope of peak 1 (.35) is within the confidence bounds of peak 2 ([.16, .40]). **b**) As in Supplementary Figure 3 (panels **h** and **i**), constant and cumulative delay were estimated by fitting a linear model to the peak data, and calculating the intercept and slope of those peaks, to estimate constant and cumulative delay respectively. Analysis of visual latencies revealed a similar pattern as the template-fitting approach, i.e., of visual constant delay in absence of visual cumulative delay. However, the variance explained by peak-fitting ($R^2=0.02$) was considerably lower than for template-fitting ($R^2=0.11$), suggesting that peak-fitting was not as sensitive. Analysis of auditory peak latencies revealed a positive effect of age on auditory cumulative delay ($R^2=0.09$), but a small negative effect of age on auditory constant delay ($R^2=0.04$). It is clear from examining plots in panel **c** that there is no negative relationship between age and constant delay in the auditory ERF (which would correspond to a negative shift of the P1m in the older subjects). This effect is likely a spurious result due to the lower sensitivity of peak fitting compared to template fitting approaches.

The new procedure used to obtain the ERP template and to calculate the latency should be presented earlier in the text.

We have moved the description of the ERF fitting earlier in the methods section, and have also improved the description to clearly state that the fitting procedure takes in to account individual differences in amplitude offset and scaling.

We have also included the following paragraph in the Results:

In order to estimate constant and cumulative delay for each participant, a template fitting procedure was employed, in which the group average signal (gray line in Fig. 1d) was fit to each participant's ERF by a combination of temporal displacement (constant delay) and temporal linear dilation (cumulative delay). Using a local gradient ascent algorithm, these two parameters were adjusted until the linear model fit (R^2) was maximised. Using R^2 as the utility function simplifies the fitting procedure, and allows simultaneous estimation of the amplitude offset and amplitude scaling (see Methods).

Moreover, it is unclear to what extent the new technique takes into account changes in ERP amplitude, which would alter the morphology and might be confounded with differences in latency. For instance, the auditory P1 wave notoriously increases in amplitude with age, but its peak shows little sensitivity to aging. Visual evoked responses show less age-related increased in amplitude. Could the age-related effects on constants and cumulative delays be partly accounted for by different patterns of age-related changes in ERP amplitude for visual and auditory stimuli? The manuscript should consider this in the discussion.

Our template-fitting procedure includes a scaling parameter to capture differences in overall amplitude (and an offset parameter to capture changes in the ERF mean). As these are orthogonal parameters of the fitting procedure, offset, and amplitude scaling can be recovered independently of delay from the intercept and slope of the linear model. Nonetheless, it is possible that the amplitude of different ERP components are differentially affected by age, and in that case could affect delay estimates; a point we now note in the Discussion. However, we do not think differences in P1 amplitude confound our latency estimates for two reasons: 1) We repeated the template fitting on a much shorter time window encompassing only the N1m, where we found largely the same effects of age on auditory cumulative and visual constant delay as when fitting the whole epoch (see Supplementary Figure 1, also shown in response to Reviewer 1); 2) If the template fitting approach were not sufficient to capture ageing effects in the ERFs, we would see a significant correlation between delay and root mean square error (RMSE) of the fit, which is not the case (Spearman's $Rho = .06$, $p > 0.05$). Furthermore, we control for RMSE when estimating age effects on our delay parameters (see Supplementary Table 1).

	Age vs. Delay	Age vs. Acuity	Delay vs. Acuity	Delay vs. Acuity Cov=Age	Age vs. Delay Cov = Acuity	N
Aud. Cumtv.	.37***	-.46***	-.22***	-.06	.31***	567
Vis. Const	.32***	.43***	.13**	-.01	.30***	511
		Age vs. Amp	Delay vs. Amp	Delay vs. Amp Cov=Age	Age vs. Delay Cov = Amp	
Aud. Cumtv.	-	.20***	.06	-.02	.36***	582

Vis. Const	.08*	-.03	-.00	.32***	511
	Age vs. Offset	Delay vs. Offset	Delay vs. Offset Cov=Age	Age vs. Delay Cov = Offset	
Aud. Cumtv.	.12*	-.15***	-.21***	.40***	582
Vis. Const	.31***	.10*	.00	.31***	511
	Age vs. RMSE	Delay vs. RMSE	Delay vs. RMSE Cov=Age	Age vs. Delay Cov = RMSE	
Aud. Cumtv.	.05	.04	.02	.37***	582
Vis. Const	-.13	-.09	-.05	.34***	511

* $P < 0.05$, ** $P < 0.01$, *** $P < 0.001$

Supplementary Table 1: Partial rank correlations (Spearman's Rho) demonstrating that the age vs. delay relationships observed in the main results section are not accounted for by visual or auditory acuity, age-related changes in amplitude scaling (Amp), amplitude offset (Offset) or root mean square error (RMSE) of fit. Column 2 shows the relationship between age and the potential confounding variable (e.g. auditory/visual acuity). Column 3 shows the relationship between evoked response delay and the confounding variable. Column 4 shows the relationship between delay and the confounding variable while controlling for age. Column 5 shows the relationship between age and delay while controlling for the confounding variable. In all cases, except for amplitude offset, controlling for age abolishes the relationship between delay and the confounding variable. In all cases, controlling for the confounding variable has very little effect on the age-delay relationship. Auditory acuity measures were not available for 15 participants, who were excluded from the above analysis.

The age-related change in latency reported in this study appears small compared to those reported for the auditory N1 and P2 wave. Again, it would be useful to show conventional measurement of the N1 and P2 wave. The peak-to-peak difference in latency can also be reported as it can be used an index of cumulative delay.

Differences in peak-to-peak latency are implicit in the linear fit to successive peak latencies in Supplementary Figures 3 and 4 above. Nonetheless, for completeness, we extracted just the N1-P2 latency, and found that this correlated with age for auditory ($R^2=0.10$, $p < 0.001$) but not visual ($R^2 < 0.01$, $p = 0.130$) responses. Nonetheless, we reiterate that this peak-to-peak latency estimate seems likely to be more prone to noise (in that age did not explain such a large proportion of variance).

The behavioral data from the attention tasks should be analyzed and reported in the text. It is important to demonstrate that the current sample shows the typical age-related increases in response time. It would also be interesting to explore the relation between changes in ERP, white and grey matter on response time.

We did not originally include RT data, because the active task was not speeded (and there are no RTs in the passive task). We now report RT correlations in Supplementary Table 2. The lack of speeded instructions is probably why the correlation with age was not significant ($R = .06$, $p > .05$). Nonetheless, for completeness, we correlated RTs with all four delay parameters from both tasks, and only the auditory constant delay parameter was significantly correlated with RTs, in both tasks, and either with or without adjustment for age. This was not a parameter that showed a significant age effect, but may relate to the general speed with

which participants responded to the tones. However, because of the unspeeded nature of this task, we do not speculate further.

	Passive			Active		
	RT	RT Cov = Age	N	RT	RT Cov = Age	N
Age	--	--	--	0.06	--	609
Aud Const.	0.15***	0.15***	567	0.10*	0.12***	535
Aud Cumtv.	-0.02	-0.05	567	0.00	-0.03	535
Vis Const.	0.05	0.03	511	0.03	0.01	545
Vis Cumtv.	0.00	0.00	511	0.03	0.04	545

* $P < 0.05$, ** $P < 0.01$, *** $P < 0.001$

Supplementary Table 2. Rank correlations (Spearman's Rho, r_s) of reaction times in the active task (RT) with age and delay estimates in both tasks (there were no RTs in passive task). Correlations were performed with and without age as a covariate. RT was not correlated with age ($p < 0.05$), likely reflecting the non-speeded nature of the active task. There was a weak positive correlation between RT auditory constant delay in both passive and active tasks. No other delay parameters were correlated with RT.

Figure 1 shows the template used to extract latencies, but this template appears to be limited to sensory evoked responses. The manuscript does not present the ERPs elicited by target stimuli during the active listening task. These are particularly important because they should be closely related to behavior.

All of the trials in the active task were target stimuli (there were no separate targets). In order to make things clearer, we have combined the previous Supplementary Figures 1 and 2 into the new Supplementary Figure 2 to show that when evoked responses from the active task are transformed using the same weights as those in the passive session, the evoked responses look very similar.

Listing "CamCAN" as an author is peculiar. I understand that the data are coming from a consortium and this is already indicated in the acknowledgement. Hence, there is no need to add the consortium as a co-author, is there? This would bring the definition of co-authorship to a new level of confusion, inconsistent with usual criteria for authorship of scientific work.

This is the standard method of citing CamCAN, and has been used before in Nature Communications (see Kievit et al. 2014 <http://www.nature.com/articles/ncomms6658>); so we defer to the decision of the journal.

In the method section, the description of the auditory and visual tasks should precede the sections on MRI and MEG. Also, event-related potential (ERP) should be replaced with event-related field (ERF). Lastly, the reference to N1 and P2 should include the lower case letter m (e.g., N1m).

The text has been modified as suggested, thank you.

The description of the visual task should include the visual angle for stimulus

presentation as well as the rise/fall time and intensity for the auditory stimuli. The inter-stimulus interval should be reported as well as the steps (e.g., stimuli were presented with ISI ranging from 500 ms and 1000 ms with 10 ms steps, rectangular distribution). The term passive task does not make much sense should be replaced with passive recording condition (i.e., no response required).

We have modified the text in the methods section to read:

The visual stimulus consisted of two circular checkerboards presented simultaneously to the left and right of a central fixation cross (34ms duration x 60 presentations). The diameter of each circular checkerboard subtended an angle of 3°, their centroids were separated by 6° on the horizontal plane, and the checks had a spatial frequency modulation of 2 cycles per degree. Visual stimuli were presented using a Panasonic PT-D7700 DLP projector (1024x768, 60Hz refresh rate) outside of the MSR, projected through a waveguide onto a back-projection screen placed 129cm from the participant's head. The stimulus onset was adjusted for the 34ms (2 refreshes at 60Hz) delay induced by the projector. The auditory stimulus was a binaural tone (300ms duration; 20 presentations of 300Hz, 600Hz, and 1200Hz; 60 total presentations), with a rise and fall time of 26ms, and presented at 75dB sound pressure level. The stimulus onset was adjusted for the 13ms delay for the sound to reach the ears. The first session involved the active task, in which participants were presented with both types of stimuli concurrently, and asked to respond by pressing a button with their right index finger after stimulus onset. The reaction times (RTs) are analysed in Supplementary Table 2, though note that the active task was not explicitly speeded. The stimulus-onset asynchrony varied randomly between 2s and 26s, to match an fMRI version of same task (see Taylor et al.⁵⁶ for more details). The session lasted 8min and 40s in total. The second session was the passive task. Participants experienced separate trials with either the visual or auditory stimulus, which they were asked to passively observe (no response required). Visual and auditory trials were pseudo-randomly ordered, with an SOA that varied randomly between 0.8s to 2s. Given the simultaneous presentation, evoked responses were adjusted by the average delay of 23ms (13ms auditory and 34ms visual). The session lasted approximately 2mins.

Line 73, I would replace "Some" with "Many." Indeed, there are many studies reporting effects of age on sensory evoked response latency, including the N1 and P2 waves. It is correct to say that few studies have reported age effects on middle latency evoked responses and that age effects on N1 latency are more equivocal with some studies reporting age-related increased in latency while other do not.

The text has been amended as suggested.

Line 152-153, it is unclear whether the ERP traces are actually brain waves or the most dominant component from the PCA. Given the arbitrary unit, one would assume the later, but this need to be clarified. Also, whenever possible avoid the use of arbitrary unit for y-axis.

All of our ERF plots are of the dominant component from the PCA. Therefore, all of our units are arbitrary, as we now state clearly at the end of the first paragraph of Results.

Line 257, delete "Finally"

Line 401, please add a comma after e.g.

Line 418, replace "are" with "were"

Amended as suggested.

Reference section: The references on frequency following responses (FFR) do not seem appropriate given that the present studies focuses on cortical evoked responses. The ref # 42 also seems out of place. I would highly recommend that the authors more carefully review prior work on aging using EEG, MEG and fMRI. Notably, there are several highly relevant MEG studies on sensory systems and aging that are currently omitted from their review of the literature.

We agree that the literature was missing some important work and we have made some efforts to correct that. Unfortunately, due to space limitations we could not provide a comprehensive overview of the substantial body of work on both visual and auditory evoked responses. Instead, we have included some specific examples that we thought were relevant to our study, and included some useful reviews for further reading. We hope this is satisfactory.

Reviewer #3 (Remarks to the Author):

This is a very interesting research project, attacking from a valid angle, and delivering compelling evidence for its arguments. Its main strength is arguably the unusually large sample size of >500.

The authors show using a arguably more robust all-timepoints-based estimation technique that auditory and visual cortical-response changes slow down differentially in older age (cross-sectionally, of course, as the authors correctly concede); suggesting different aging mechanisms behind cumulative and constant response delay, respectively.

They link these cortical slowings in interesting, and still relatively simple (i.e., parsimonious) mediating-analysis models, to brain structural data and are thus able to link changes in certain brain areas.

I am very much in favour of this manuscript, and very much enjoyed reading it. I offer comments below that the authors should find helpful in re-considering some aspects of their analysis and presentation. They mostly relate to aspects of model fitting and statistics.

We thank the reviewer for their positive comments, and the time they devoted to our submission.

-- A corollary from the cumulative-delay idea: Should the first indication of differential age-shift in audio vs visual not be nicely visible if the ERO in Fig 1d or elsewhere would show some measure of dispersion? I would expect that the dispersion (while increasing probably for both, aud and vis, with time) increases more drastically with time in audition.

Anyhow, a way to have Fig. 1d reflect the effect would be very welcome.

We have plotted ERFs averaged within three age groups in Supplementary Figure 1 above (in response to Reviewer 1's first comment), where dispersion of components with age can indeed be seen clearly in the auditory responses. We also conducted an additional analysis restricted to the early parts of auditory and visual responses to ensure that our effects were not due to other types of changes in the morphology of the waveform. For example, in the auditory response, it is not clear from inspection of the ERFs for the three age groups whether a true cumulative delay (dispersion) exists in both early and late components, and whether the late component is shifted independently of the early component. Nonetheless, even when analysing just the first 0-140ms, we find an effect of age on cumulative but not constant delay, suggesting the effect is not driven solely by the later, broad components.

Similarly in the visual ERF, an amplitude change in the early component raises the concern about whether there is a negative cumulative delay (i.e, contraction) of the first peak, or a change in constant delay without a contraction. Our additional analysis on just the early timewindow (0-140ms) confirms that a constant delay is observed in absence of any contraction of the waveform.

-- This bring me to one of my more major methodological concerns: Is the fitting method, as intriguing as it sounds, really fair? Could it be improved by using some form of cross-validation/train-and-test approach? As I understand it, the average ERP across all subjects was first derived and used as a template. Now, this average ignores the

dispersion that (we know from the rest of the paper) is increasing with time in audition (but not in vision), because of age (for early time points, young and old listeners have the "same" ERP at a given time point, for later time points they increasingly diverge). We could thus assume that later portions of the (particularly auditory) average ERP are a worse representation of the individual ERP, and this might or might not bias/influence parameter estimates and R^2 .

Maybe the authors have a good reply to this at hand, but maybe an approach where the template is not biased towards the hypothesis (worse fit later on, thus maybe favouring one or the other model) might be preferable? A cross-validation approach, where the template is somehow derived independent of the to-be-fitted data, might help.

Firstly, we do not think there is any simple bias in using the averaged template across subjects to then test for age effects – the mean and slope of a regression are independent – so cross-validation seems unnecessary.

However, we accept that it may be possible that a selective age effect on one part of the ERF might bias the template. So we have conducted further analyses to show that using either a template defined solely from young participants, or solely from old participants, yields very similar results. These results are now presented in Supplementary Figure 7 (reproduced below).

Supplementary Figure 7. Effect of using different templates on the fitting procedure. As expected, using a template derived from young participants (<30ys) resulted in a shift of the constant and cumulative delay estimates in the positive direction. Similarly, using a template from older subjects only (>60ys) resulted in a negative shift of delay estimates. Nonetheless, the pattern of significant age effects was identical to that when using a template averaged across all participants, i.e., on visual constant delay and auditory cumulative delay.

-- The one thing that this paper lacks in order to be of markedly higher relevance would be a plausible (computational) model of how the differential effects of age on auditory and visual responses can come about. This is asking for a lot, I know; but given the [senior] authors' experience in this field, I was expecting something like this.

We agree that computational modelling would provide a more comprehensive explanation as to how constant and cumulative delay can be dissociated. We have thought about this, but are not sure there are sufficient constraints from the present analyses to produce a model that goes much beyond our current verbal account. Nonetheless, we do plan to explore these data further in future, for example using neurophysiological models (like the neural mass models used in Dynamic Causal Modelling of evoked responses, for example) that have been tested in other contexts. We have amended the manuscript to explain that this is an important next step.

The section of the neurobiology behind these effects had to remain speculative, and I am not convinced why

(i) auditory responses should behave so differently from visual responses; the most marked difference lies in the number of synapses and complexity of wiring *before* the afferent signal reaches cortex, but the authors seem to imply/observe that ... (ii) age-related delay should only start in cortex. There are by now many studies linking or trying to link ageing to alterations in the sensorineural periphery, especially in the auditory domain (often or mostly employing brainstem responses, frequency-following response, e.g. from the Kraus lab, the Shinn-Cunningham lab, Terry Picton and others). Why should the neurobiological explanation offered here, e.g. altered feedback/inhibition between STS and HG, apply not earlier in the processing hierarchy?

In response to comments, we have made efforts to include a broader range of literature in both introduction and discussion. However, we did not have a clear hypothesis of which regions of the brain should mediate neural delay, and therefore the study was somewhat exploratory. The fact that we performed a whole-brain grey and white matter mediation analysis reflects that. Our interpretation, that cumulative delay was based on interactions between grey matter regions was based mainly on the results of that whole-brain analysis. It is of course possible that delays also exist in peripheral or central auditory systems, but we did not observe any evidence for that. We simply assume that if there were significant delays in peripheral system, it would reflect as a constant delay in the early parts of the evoked response (as it does in visual response). For example, in the (relatively long) visual pathway, changes in white matter connections mediate changes in constant delay. Therefore, it seemed that the idea of white matter atrophy resulting in a delay in the arrival time of information to the cortex is the most plausible and likely explanation. However, the mediation results for auditory cumulative delay and grey matter atrophy are harder to interpret, and we simply propose that the mechanisms involved here might result from changes in local computation within cortical regions, as opposed to simple delays in one-way transmission along white matter pathways between regions. We do now acknowledge in the discussion that amplitude increases may be the result of disinhibition of the central auditory system (in agreement with the review by Tremblay and Ross, 2007), and we include this as a reference in our caveats. Unfortunately, due to the limitations of our non-invasive neuroimaging methods, we do not feel that we can speculate further on specific mechanisms.

I would like to see the authors acknowledge more upfront the (modest) effect size of their main findings. They do have the numbers to crunch out statistical effects. But this does not change the fact that an R^2 of .1 is only 10% of the variance explained. (I agree that this is noteworthy, but it is also clear that in more conventional sample sizes the same effect, an r of .3, would hardly pass conventional levels of significance and thus go often unnoticed in the literature.)

We now acknowledge that the effect sizes are low, and perhaps these effects would not be visible in smaller sample sizes. To aid the reader in determining the probability of finding a significant age-effect in a smaller sample, we have performed a power calculation illustrated in Supplementary Figure 5.

Supplementary Figure 5: Non-parametric power calculations obtained by bootstrapped re-sampling, from $N=10$ to the full sample here, in steps of 10, with 10,000 re-samples per sub-group size. The blue line indicates the power (the probability of rejecting the null hypothesis at the $p<0.05$ level, given the alternative hypothesis is true). Therefore, P is calculated as the proportion of resamples in which the lower confidence interval was greater than 0.

Related, they speak of "mediation effect sizes [per voxel]". Where are they reported/what are they used for? A p-value of course is *not* an effect size, but I am uncertain what the authors refer to here.

Effect sizes are reported in the statistical maps as percentage of covariance (between X and Y) explained by the mediating variable (M). We have modified our plots and hopefully this is now clearer (see Figure 3).

-- I am not convinced by using linear (robust) regression methods on variables that are clearly, by their range of values, not normal. (See e.g. Sensory Acuity, bound to $[-\infty; 1]$, or percentages like visual amplitude, bound to $[0;\infty]$). The authors should apply adequate transforms (log, logit, arcsine, etc.) or apply rank regression. This might change conclusions in particular in some voxels, where relations now thought to be meaningfully linear turn out not to be and vice versa.

The General Linear Model (GLM) that underlies all our parametric tests does not assume that the regressors (independent variables) are normally-distributed. It only assumes that the error is normal, as estimated from the residuals. We did not find any visual evidence of non-normal residuals when variables like acuity or amplitude were used as regressors of no interest.

We accept however that when such variables are the dependent variable in the GLM, a non-normal distribution of such data is likely to lead to non-normal residuals. Fortunately, there was no evidence that our delay estimates were not normally-distributed after removing outliers (Kolmogorov–Smirnov tests were non-significant in all cases), so we did not apply any transforms to these variables. In the case of acuity, amplitude and RMSE of the fit, we re-analysed the data using partial rank correlation as suggested. Our conclusion, that delay parameters are not affected by acuity, signal amplitude or fitting error, remained the same. Now that those relationships are analysed using Spearman correlations, we have removed supplementary figures 3, 4 and 5 and replaced them with a table, which is hopefully more concise. For completeness, we also included ERF offset as an additional covariate, since this parameter may also explain some of the age-related amplitude differences across age.

-- It speaks in favour of the authors that they resisted the temptation to posthoc “spin” detailed hypotheses about which brain structures (white and/or grey matter) would potentially influence cortical slowing -- but the approach nevertheless leaves us with a "fishing expedition" (albeit with adequate statistical control of type I error), so the brain areas identified and discussed might be taken by some readers with a grain of salt.

We have added a note to this effect in the General Discussion under caveats.

-- very minor: Some figures, at least figure 1, contain a somewhat ugly mixture of typefaces, Arial and Calibri.

This has been amended as suggested – thank you.

REVIEWERS' COMMENTS:

Reviewer #1 (Remarks to the Author):

The authors have addressed all my questions and I strongly endorse the publication of their article. The new figures and the revised text are all excellent. Figure 3 is a much better representation of the results and I'm sure to use it as an example to follow in my next reviews. I greatly value all the time and effort that went into revising the paper and addressing questions from the 3 reviewers. The reviewers' comments and the authors' replies are a fascinating read and should really be published alongside the paper. The technical discussions will be of interest to a large readership and will demonstrate that this field is moving on, embracing new techniques and a more thorough computational approach. To me this is a landmark paper, and all the more exciting because we do not really understand the results yet... As noted by one reviewer, the effect sizes are relatively small. But to be fair, most ageing studies compare two groups and focus mostly on whether the group difference is statistically significant, without ever reporting the R^2 of their ANOVA. I bet the majority of ageing effects explain very little variance, and these effects are inflated because of small sample size and publication bias. So the present study shines because of its large sample size and systematic quantification of the effects in two tasks and two modalities.

I've just noticed a few typos:

Supplementary Figure 3

"Peak latencies alone however are affected by both constant and cumulative delay, the influence of each depending on the time point under observation." depending -> depends

Supplementary Figure 5

"the blue line indicates the power" - in that figure both blue and red lines indicate power.

Conclusion

"the present work fills a missing gap in the literature" - missing is not needed

Reviewer #2 (Remarks to the Author):

The authors have done an excellent job at revising their manuscript. They have answers and address all my prior comments. I do not have further comments and I would like to congratulate the authors for an excellent and likely very influential study.

Reviewer #3 (Remarks to the Author):

The authors must be congratulated on making a really compelling, large-scale study even more attractive and scientifically sound, by taking serious my and the fellow reviewers' manifold concerns. I am very content with the answers I received, and I learned a lot from studying the response letter.

I have no objections to this revised paper appearing as soon as possible in Nat Comms.

Jonas Obleser

PS: If I may, I would only like to join Reviewer 1 in his objection of having the CamCan collective as an author identity, and would like to bring this to the editors' attention (to whom the authors in their response have deferred the decision) here one more time.

REVIEWERS' COMMENTS:

We would like to thank all reviewers for dedicating their time to reviewing this paper. We feel that their constructive comments have greatly improved the manuscript.

Reviewer #1 (Remarks to the Author):

The authors have addressed all my questions and I strongly endorse the publication of their article. The new figures and the revised text are all excellent. Figure 3 is a much better representation of the results and I'm sure to use it as an example to follow in my next reviews. I greatly value all the time and effort that went into revising the paper and addressing questions from the 3 reviewers. The reviewers' comments and the authors' replies are a fascinating read and should really be published alongside the paper. The technical discussions will be of interest to a large readership and will demonstrate that this field is moving on, embracing new techniques and a more thorough computational approach. To me this is a landmark paper, and all the more exciting because we do not really understand the results yet... As noted by one reviewer, the effect sizes are relatively small. But to be fair, most ageing studies compare two groups and focus mostly on whether the group difference is statistically significant, without ever reporting the R^2 of their ANOVA. I bet the majority of ageing effects explain very little variance, and these effects are inflated because of small sample size and publication bias. So the present study shines because of its large sample size and systematic quantification of the effects in two tasks and two modalities.

We thank Dr. Rousselet for his positive review, and we have informed the Editor of our intent to publish reviewers' comments alongside the published paper.

I've just noticed a few typos:

Supplementary Figure 3

"Peak latencies alone however are affected by both constant and cumulative delay, the influence of each depending on the time point under observation." depending -> depends

This has been amended as suggested

Supplementary Figure 5

"the blue line indicates the power" - in that figure both blue and red lines indicate power.

This has been amended to "Blue and orange lines indicate the power"

Conclusion

"the present work fills a missing gap in the literature" - missing is not needed

This has been amended as suggested

Reviewer #2 (Remarks to the Author):

The authors have done an excellent job at revising their manuscript. They have answers and address all my prior comments. I do not have further comments and I would like to congratulate the authors for an excellent and likely very influential study.

We thank reviewer 2 for their positive review and for all their hard work reviewing the manuscript.

Reviewer #3 (Remarks to the Author):

The authors must be congratulated on making a really compelling, large-scale study even more attractive and scientifically sound, by taking serious my and the fellow reviewers' manifold concerns. I am very content with the answers I received, and I learned a lot from studying the response letter. I have no objections to this revised paper appearing as soon as possible in Nat Comms.

We thank Professor Obleser for his positive review and helpful comments on our manuscript.

Jonas Obleser